# Universal geometric frustration in pyrochlores

B.A. Trump[1,2,3], S.M. Koohpayeh[3], K.J.T. Livi[4], J.-J. Wen[3], K.E. Arpino[2,3], Q.M. Ramasse[5], R. Brydson[6], M. Feygenson[7], H. Takeda[8], M. Takigawa[8], K. Kimura[9], S. Nakatsuji [8], C.L. Broholm[3,4] & T.M. McQueen[2,3,4]

Materials with the pyrochlore/fluorite structure have diverse technological applications, from magnetism to nuclear waste disposal. Here we report the observation of structural instability present in the pyrochlores $A_2Zr_2O_6O'$ ($A$ = Pr, La) and $Yb_2Ti_2O_6O'$, that exists despite ideal stoichiometry, ideal cation-ordering, the absence of lone pair effects, and a lack of magnetic order. Though these materials appear to have good long-range order, local structure probes find displacements, of the order of 0.01 nm, within the pyrochlore framework. The pattern of displacements of the $A_2O'$ sublattice mimics the entropically-driven fluxional motions characteristic of and well-known in the silica mineral β-cristobalite. The universality of such displacements within the pyrochlore structure adds to the known structural diversity and explains the extreme sensitivity to composition found in quantum spin ices and the lack of ferroelectric behavior in pyrochlores.

[1] NIST Center for Neutron Research, National Institute of Standards and Technology, Gaithersburg, MD 20899, USA. [2] Department of Chemistry, Johns Hopkins University, Baltimore, MD 21218, USA. [3] Department of Physics and Astronomy, Institute for Quantum Matter, Johns Hopkins University, Baltimore, MD 21218, USA. [4] Department of Materials Science and Engineering, Johns Hopkins University, Baltimore, MD 21218, USA. [5] SuperSTEM Laboratory, STFC Daresbury Campus, Daresbury WA4 4AD, UK. [6] School of Chemical and Process Engineering, University of Leeds, Leeds LS2 9JT, UK. [7] Jülich Center for Neutron Science, Forschungszentrum Jülich GmbH, D-52425 Jülich, Germany. [8] Institute for Solid State Physics, University of Tokyo, Kashiwa, Chiba 277-8581, Japan. [9] Division of Materials Physics, Graduate School of Engineering Science, Osaka University, Toyonaka, Osaka 560-8531, Japan. Correspondence and requests for materials should be addressed to T.M.M. (email: mcqueen@jhu.edu)

From catalysis, ferroelectricity, luminescence, magnetism, and even nuclear waste storage, compounds which crystallize in the pyrochlore structure are utilized for countless applications[1–3]. This makes knowledge of the structural flexibility of the pyrochlore structure, $A_2B_2O_6O'O''_0$, a derivative of the fluorite structure, crucial. The structure contains interpenetrating, corner-sharing $O'A_4$ and $O''B_4$ tetrahedra, shown in Fig. 1a. $A$ is typically a larger (~0.1 nm) cation while $B$ is a smaller cation (~0.05 nm), where the central $O''$ site is unoccupied.

Stoichiometric flexibility is common for the pyrochlore structure, such as in $Y_4Nb_3O_{12}$ ($Y_2(Y_{0.14}Nb_{0.86})_2O_{6.91}$) which contains both an $A$ cation on the $B$ site and O vacancies[4]. Additional disorder, such as $A/B$ site-mixing is not uncommon[5–8], and can lead to the weberite structure[9], unless optimal cation ordering occurs. Lastly, polyhedral distortions can also occur, where the tetrahedral corners move in or out. If in/out displacements have long-range order then the global symmetry drops from $Fd\bar{3}m$ to $F\bar{4}3m$ evidenced by the addition of $2h\,0\,0$ reflections, as seen for $Pb_2Ru_2O_{6.5}$, due to a combination of lone pair effects and off-stoichiometry[10]. If in-out ordering only exists over short-length scales, it is equivalent to 2-in/2-out (2I2O) ordering, which is expected to be disordered on long-length scales due to degenerate configurations[11]. This is observed for Nb-pyrochlores due to charge disproportionation[4], and also for $Y_2Mo_2O_7$ observed as split peaks in X-ray absorption spectroscopy[12]. Though this disorder is often not magnetically driven, for pyrochlores with magnetic cations, it is this same short-range order with degenerate configurations which leads to non-trivial magnetic states[13, 14]. Furthermore, understanding the magnitude and direction of these displacements elucidates next-nearest neighbor magnetic cation distances, which are especially important for pyrochlores with quantum magnetism.

An alternative type of disorder, cooperative tetrahedral tilting, also exists for compounds such as $Bi_2Ti_2O_7$ and $Bi_2Ru_2O_7$, and is attributed to lone-pair effects[15–17]. Ferroelectric behavior is expected from this disorder, though these materials are only dielectrics[16, 18], with ferroelectric behavior attributed to impurities[19]. Better understanding this type of disorder is essential to describe why these pyrochlores are not ferroelectric, despite the local structure deviating from cubic symmetry, as well as understanding why so few pyrochlores are ferroelectrics despite having large dielectric constants[1].

Here we provide evidence for this disorder in the pyrochlores $Pr_2Zr_2O_7$ (PZO), $La_2Zr_2O_7$ (LZO), and $Yb_2Ti_2O_7$ (YTO), due to static $A$ and O displacements (see Fig. 1c). These displacements mimic isoreticular β-cristobalite, where corner-sharing tetrahedra cooperatively tilt, allowing for an increase in the Si–O bond length and a deviation of the O–Si–O angle from 180° (see Fig. 1d)[20]. The displacements lower the local symmetry to $P4_32_12$ and exist here despite ideal cation ordering, ideal stoichiometry, the absence of lone pair effects, and despite LZO being nonmagnetic. Powder diffraction and selected area electron diffraction (SAED) hint at structural disorder, where pair-distribution function (PDF) analysis refinements prefer static, rather than thermally dynamic, displacements. Additionally, $^{91}Zr$ NQR on a PZO series indicate an inhomogeneous bonding environment of Zr as a function of $A/B$ cation ratio, while high-angle annular dark field scanning transmission electron microscopy (HAADF STEM) reveals a static displacement of ~0.01 nm for $A$ and O. Though previous works have observed similar displacements and the associated structural frustration[21, 22], we generalize that these static displacements are common in pyrochlore structures and are driven by cation size mismatch, rather than defect concentrations or electronic effects.

## Results

**Diffraction data.** Rietveld refinements of powder X-ray and neutron diffraction on a single crystal of LZO display neither the {200} reflection nor any secondary phases (see Supplementary Figure 1). Refinements used anisotropic displacement parameters (ADP), yielding anisotropic displacement ellipsoids (ADE) such as those shown in Fig. 1b. Details of X-ray diffraction on PZO and YTO are found in refs. [23, 24] respectively. All $A$-site cations have pancake-like ADE, suggesting tetrahedron tilting, rather than the rod-like ADE indicative of in/out ordering. Systematic refinement tests were conducted for PZO, LZO, and YTO for off-stoichiometry, cation site-mixing, or O off-stoichiometry and each test indicated the absence of these effects (see Supplementary Discussion for more details). This allows us to conclude that the stoichiometry and occupancies are within 0.5% of nominal values (see Supplementary Figures 2–28), and indicates unambiguously that these materials are nearly defect free.

Additional tests were conducted for evidence of structural displacements, by placing the $A$ and $B$ cations on the $32e:(x\,x\,x)$, $96h:(0\,y\,-y)$ and $96g:(z\,x\,x)$ Wyckoff positions. Pr, Zr, La, and Yb all showed an equal preference for the $96g$ and $96h$ sites, and Ti showed only a preference for the $32e$ site. To obtain reliable information about O displacements, similar tests were conducted using neutron diffraction data, confirming that La and Zr prefer the $96g/96h$ sites, O prefers the $96g$ site, and O' does not move. Though alternative Wyckoff positions (static ordering) improve refinement statistics, this improvement was not statistically significant from fits with ADP (dynamic ordering). This analysis merely suggests displacements are static, which would have significant impacts on both magnetic and ferroelectric behavior.

Intensity on the allowed {442} reflection would suggest a static β-cristobalite disorder due to $A$, $B$, or O displacements[25]. Although none of the powder diffraction data have intensity on the {442} reflection, Supplementary Figure 1a demonstrates that

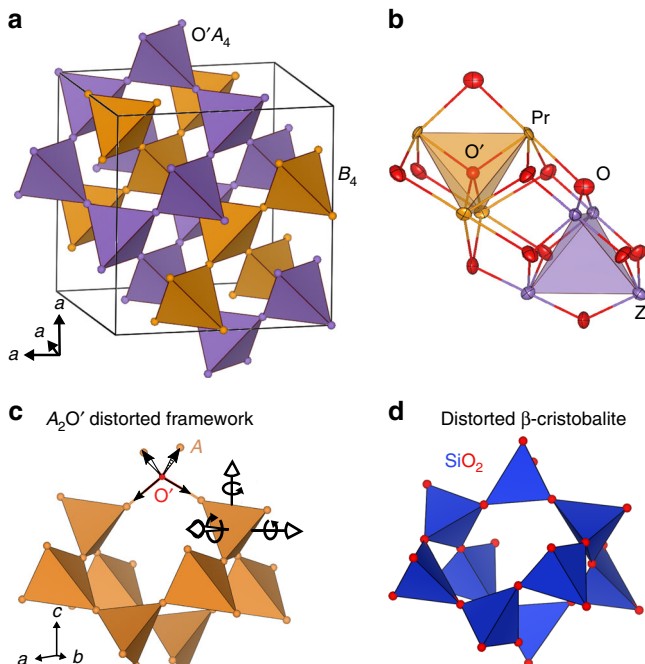

**Fig. 1** The pyrochlore structure and structural flexibility. **a** The $A_2B_2O_6O'$ pyrochlore lattice shown as interpenetrating, corner-sharing $O'A_4$ and $B_4$ tetrahedra, with O omitted for clarity. **b** A section of the pyrochlore structure, highlighting the O connectivity. Pr (orange), Zr (purple), O (red), and O' (red) are shown as anisotropic displacement ellipsoids from NPDF refinements. **c** Cooperatively tilted $O'A_4$ tetrahedra in the $A_2O'$ network of the pyrochlore structure allow for $A$-O' bonds to lengthen. **d** A proposed structure for isoreticular β-cristobalite, which contains larger displacements and is more disordered

intensity on the {442} reflection would only be noticeable when a displacement $\delta > 0.02$ nm is present, while the displacements we measure are only $\leq 0.01$ nm (see Supplementary Figures 8, 10, 12, 19, 26, 27). Even for synchrotron data, a 0.01 nm Pr displacement produces {442} intensity that is within the error of the measurement. Similarly, the {442} reflection was not observed for neutron diffraction, though a displacement $\delta > 0.03$ nm would be needed to observe it (see Supplementary Figure 1b), making it possible that static displacements could occur despite lacking {442} intensity.

SAED on PZO and YTO crystals, oriented in the [111] direction, further complicate this analysis, as both show diffuse scattering perpendicular to the <110> directions. This is observed both in off-center SAED in Fig. 2, and centered, over exposed films, in Supplementary Figures 42, 43. In order to illuminate extremely weak diffuse scattering features, it was necessary to both over-expose and collect patterns slightly off-axis, approaches commonly used to emphasize diffuse scattering[4, 26]. This required collecting the SAED patterns in Fig. 2 off-center of the main reflections, to avoid irreversibly damaging the detector. Due to this, reflections from the First Order Laue Zone are observed as well. This pattern of weak diffuse streaks is associated with β-cristobalite disorder[27], due to $SiO_4$ tetrahedral rotations and Si–O bond lengthening. The same diffuse scattering has also previously been observed for LZO, and was hypothesized to be due to disordered static displacements which alleviate La–O′ over bonding[22, 28]. However, the diffuse scattering could indicate preferential dynamic motion instead. In comparison with previous works, our materials have ideal stoichiometry, are nearly defect free, and the magnitude of the displacements appears smaller, hence harder to observe, and thus required longer exposure times even though faint diffuse scattering exists at shorter exposure times (see Supplementary Figures 42, 43). If the displacements are static, it implies local disorder, suggesting frustrated, rather than cooperative, tetrahedral tilting, which would perturb long-range magnetic or ferroelectric order.

**Local structure probes**. To further understand the local structure, and hence the effects on physical properties, neutron PDF (NPDF) was collected on PZO and X-ray PDF (XPDF) data was collected on YTO (see Fig. 3). Fits with purely isotropic displacement parameters were not sufficient to describe the data, while fits with ADP better describe the data (see Fig. 3a, d – $R_w = 11.70$ vs. $R_w = 9.09$ and $R_w = 10.50$ vs. $R_w = 9.53$), but are still imperfect, especially at short distances ($r < 0.35$ nm).

Systematic refinement tests for alternative Wyckoff positions yielded 0.013(2) nm $96g/96h$ Pr displacements, 0.004(2) nm $96g/96h$ Zr displacements, 0.010(3) nm $96g$ O displacements, and a robust O′ position for PZO; also, yielding 0.0098(9) nm Yb $96g/96h$ displacements, 0.011(3) nm Ti $32e$ displacements, 0.015(4) nm $96g$ O displacements, and a robust O′ position for YTO (see Supplementary Figures 29, 41). These displacements are consistent with a β-cristobalite distortion, modeled in Fig. 3b, c, e, f with refinements in $P4_32_12$ symmetry (details in Supplementary Tables 1, 2).

A $P4_32_12$ refinement improves fit statistics ($R_w = 9.09$ vs. $R_w = 8.76$ for PZO and $R_w = 9.53$ vs. $R_w = 8.68$ for YTO), and visual inspection of Fig. 3b, e shows improvements in the region of the A–O′ and A–O bond lengths ($r < 0.35$ nm). Refinements using a range of $Q_{max}$ values produced equivalent fits, and refinements with similar degrees of freedom also preferred a static $P4_32_12$ model. These lower symmetry refinements, which are quantitatively and qualitatively preferred, model only several A–O′/A–O distances, rather than a large collection of distances modeled by higher symmetry refinements with ADPs. However, Fig. 3c, f demonstrate that this static model remains imperfect from

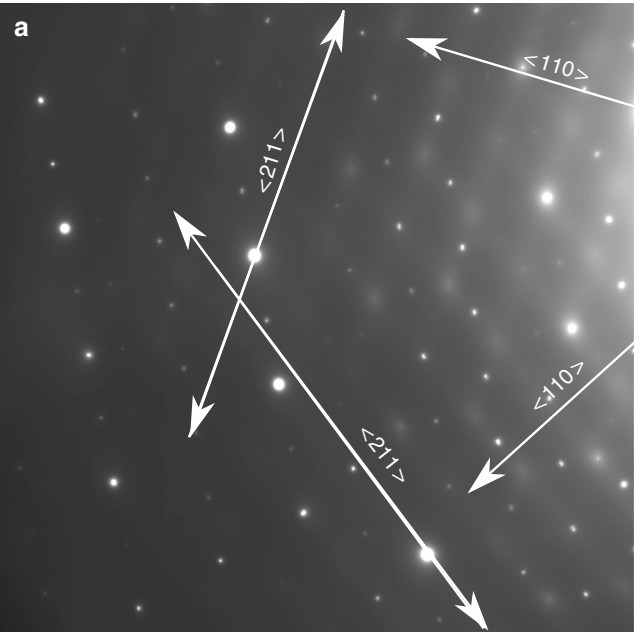

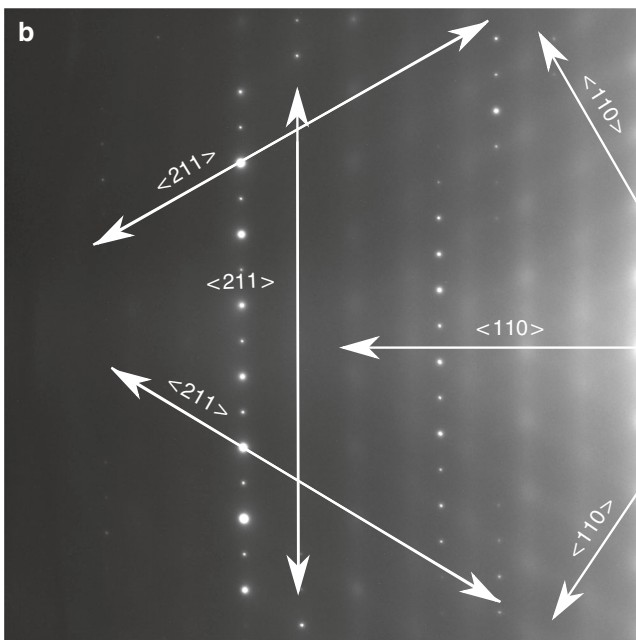

**Fig. 2** Selected area electron diffraction displaying diffuse scattering for $Pr_2Zr_2O_7$ and $Yb_2Ti_2O_7$. Single crystals oriented in the [111] direction for **a** $Pr_2Zr_2O_7$ and **b** $Yb_2Ti_2O_7$. Diffuse scattering observed perpendicular to the <110> directions. Images taken near the First Order Laue zone

0.35 nm $< r <$ 0.60 nm, and only slightly improves modeling at longer distances ($r > 0.60$ nm). This implies that the local symmetry is lowered due to static displacements in each O′$A_4$ tetrahedra ($r < 0.35$ nm), more disorder exists between tetrahedra (0.35 nm $< r <$ 0.60 nm), and that a locally disordered model accurately describes the long-range ordered structure. Further, the increased A–O′/A–O bond distances which lower the local symmetry, would suggest frustrated, rather than cooperative, motion between tetrahedra, interrupting any long-range ordered states.

$^{91}$Zr NQR is shown in Fig. 4 for a series of sintered powder PZO samples. Only one peak is observed, as expected for the ideal structure, though the breadth of the peak increases systematically as $x$ decreases in $Pr_{2+x}Zr_{2-x}O_{7-x/2}$. The broad

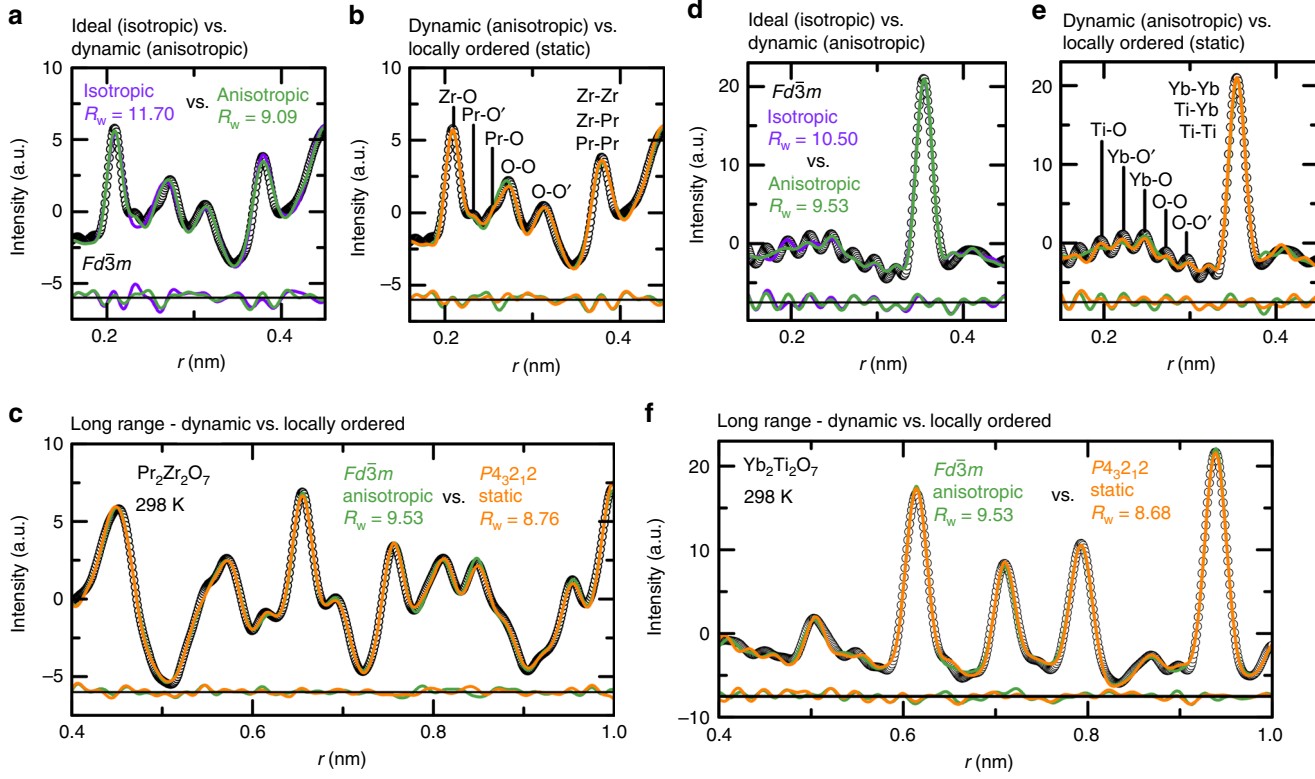

**Fig. 3** Neutron and X-ray Pair Distribution Function analyses on $Pr_2Zr_2O_7$ and $Yb_2Ti_2O_7$, respectively. Time-of-flight neutron PDF for $Pr_2Zr_2O_7$ refinements focusing on **a** and **b** short length scales and **c** longer length scales. Synchrotron X-ray PDF for $Yb_2Ti_2O_7$ focusing on **d** and **e** short length scales and **f** longer length scales. Refinements for the ideal model (isotropic displacement parameters) are shown in purple, dynamic displacements (anisotropic displacement parameters) are shown in green, and lower symmetry $P4_32_12$, locally ordered displacements (static) are shown in orange. Difference curves match their respective refinement colors and data is shown as black circles

and non-symmetric peak shape indicates that the electric field gradient (EFG) around $^{91}Zr$ is not homogeneous for $x = 0$ and $x = -0.02$. This behavior is only expected if the compound has discrete structural displacements (slower than $\approx 10^{-7}$ s, from Fig. 4) and the three-fold rotation symmetry around the [111] crystallographic axis is locally perturbed. Notably, the same asymmetric peak broadening has been observed for $^{89}Y$ NMR on $Y_2Mo_2O_7$, attributed to inequivalent Y sites at lower temperature[29]. Given that observation, peak asymmetry is expected when $x = -0.02$, as some Zr is on the Pr site would have a drastically different bonding environment. When $x = 0$ and the material is stoichiometric, the EFG for Zr is still not symmetric, suggesting non-uniform next-nearest neighbor distances. Finally, when $x = +0.02$, the Zr bonding environment appears more uniform, indicating that the EFG around $^{91}Zr$ becomes more homogeneous as the larger Pr is placed on the smaller Zr site, suggesting effects of the $A/B$ cation size ratio.

If these displacements are static, then a shift from the $Fd\bar{3}m$ Wyckoff positions should be observable in HAADF STEM. If the displacements are purely dynamic, then the average position should remain the same even if the dynamic motion is anisotropic. In contrast to the timescales due to atomic interactions ($\approx 10^{-12}-10^{-15}$ s), HAADF STEM is a long timescale measurement, of the order of seconds, much longer than the timescale expected for dynamic motion (at most $\approx 10^{-6}-10^{-12}$ s). The longer timescale and increased resolution makes HAADF STEM ideal for distinguishing between fluctuations around an ideal position (dynamic) and averaged positions that are displaced from the ideal (static).

Figure 5a shows a HAADF STEM image for PZO oriented in the [110] direction. During data collection the structure appeared

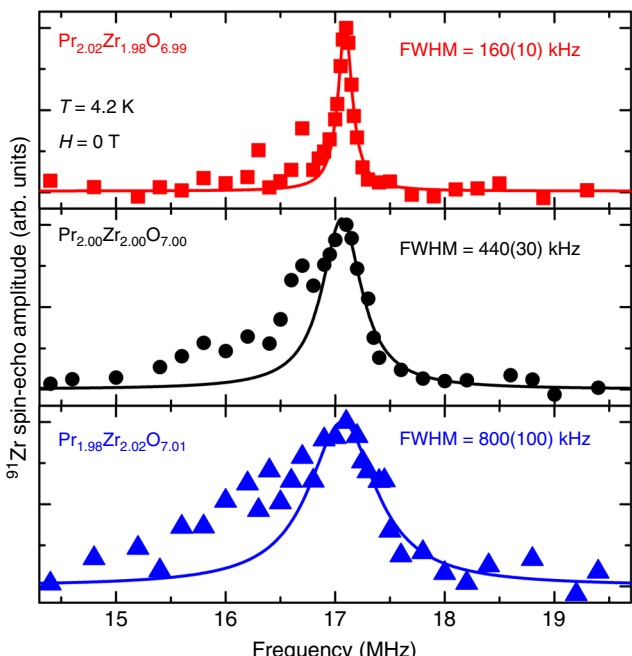

**Fig. 4** $^{91}Zr$ Nuclear Quadrupole Resonance data for the series $Pr_{2+x}$ $Zr_{2-x}O_{7-x/2}$. Experimental data is shown as red squares ($x = 0.02$), black circles ($x = 0$), and blue triangles ($x = -0.02$). Solid lines represent Lorentzian fits with respective full-width half maxima (FWHM) shown. Lorentzian fits are poorer for $x = 0$ and $x = -0.02$ due to asymmetric peak shapes. Size of the symbols represents standard uncertainties given by the scatter of the baseline

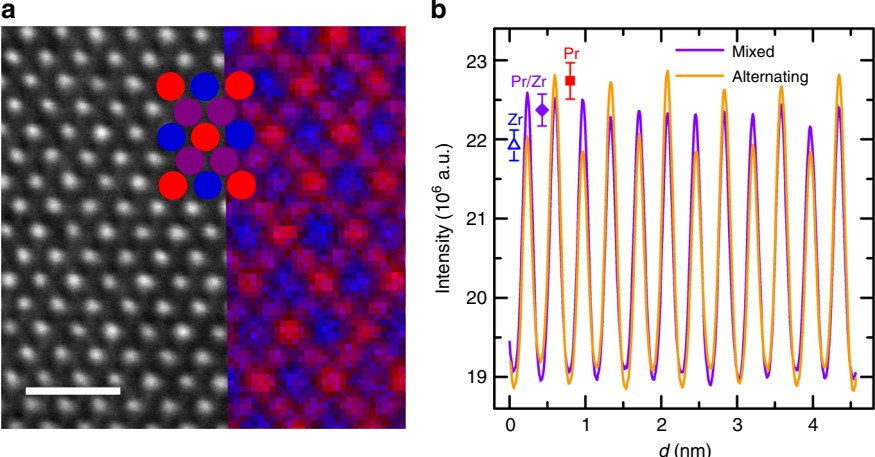

**Fig. 5** Electron energy loss spectra for $Pr_2Zr_2O_7$. **a** HAADF STEM of $Pr_2Zr_2O_7$ oriented down the [110] direction. Corresponding EELS map is shown on the right. Circles represent columns of Pr (red), Zr (blue), and mixed (purple). Scale bar represents a distance of 1.0 nm. **b** Averaged intensity profiles of PZO down the [110] direction with purple diamond, blue triangle, and red square representing average column intensity maxima for mixed columns, Zr columns, and Pr columns respectively

regular and no noticeable defects (including anti-site phase boundaries) were observed, as exemplified by representative zoomed out images (Supplementary Figures 44, 45). Figure 5a shows a corresponding electron energy loss spectroscopy (EELS) elemental map where red represents Pr, blue represents Zr, and purple represents mixed columns (see Methods section for processing details, and Supplementary Figures 46, 47 for representative images). The contrast of this map across the field of view is in close agreement with the expected A/B cation ordering, as indicated from Rietveld refinement tests. Determining precise atomic column composition from on-axis EELS data is known to be challenging due to beam propagation effects and the quantification of atomically-resolved EELS data would normally require a careful comparison with inelastic image simulations[30] or a relatively complex inversion process[31]. Nevertheless, in contrast to a recent report for off-stoichiometric $Yb_2Ti_2O_7$ samples where significant site mixing could be observed visually even in unprocessed maps[32], the uniformity of the map shown in Fig. 5a seems to be in close agreement with the lack of chemical disorder.

Rather than attempting to quantify the EELS results in terms of column composition, intensity profiles from HAADF images were instead examined to attempt to estimate the uniformity of the observed Z-contrast. Figure 5b displays the average profile intensities for each of the two row types (mixed and alternating columns) of PZO oriented in the [110] direction, with the individual row intensities displayed in Supplementary Figure 48. Given that powder diffraction and PDF refinements indicate no site-mixing or off-stoichiometry, we assume that site mixing is a negligible contribution in rows with mixed cations columns; instead, the standard deviation of mixed column rows accounts for experimental deviations of intensity (due to counting statistics). Figure 5b demonstrates that there is no significant deviation from ideal cation ordering, as the average profile intensity maxima all fall within one standard deviation of each other. Analyzing the individual row intensities, it is seen that only 0.83% (1/120) atomic columns deviate from ideal cation ordering. Using the proposed binomial distribution model[32], even assuming this column contains two defects, only suggests a stuffing defect concentration of 0.11%, well below the 0.5% limit as indicated by powder diffraction and PDF refinements (see SI for details), further indicating that the crystals are nearly defect free.

To look for static displacements of the order of 0.01 nm, a high precision workflow (see SI for details) was used to acquire representative HAADF images, following which the script Ranger was used to identify the coordinates of the atomic columns from the resulting high-resolution scanning transmission electron microscopy (HRSTEM) images[33], while ideal atomic positions were calculated from the $Fd\bar{3}m$ structure for comparison. Figure 6a shows the results of this analysis on PZO oriented in a [111] direction, with displacement vectors (black arrows) shown between ideal (black circles) and experimental (red triangles) positions. Utilizing the difference between a sub-pixel 2-D Gaussian refined experimental positions[33] and an ideal pyrochlore structure, the averaged displacements of [111] atomic columns are observed to be 0.013(6) nm, which are markedly larger than the precision typically achieved with this workflow (consecutive acquisition followed by rigid or non-rigid registration) with this instrument[34], and is in phenomenal agreement with NPDF refinements (0.013(2) nm). The displacements in the [111] orientation appear to be in a moderate agreement with those expected from the PZO NPDF refinements (overlaid on Fig. 6a), while agreement in the [110] direction is poorer (see Supplementary Figure 49).

The larger mismatch between the static $P4_32_12$ model and PZO [110] STEM image (see Supplementary Figure 49a) is not unexpected given the disorder indicated by the SAED patterns in Fig. 2. Along the [110] direction the atomic columns highlight the corners of the tetrahedra, whose positions are driven by the distortion in neighboring tetrahedra. The low connectivity, combined with competing degenerate β-cristobalite displacements, creates geometric frustration, which leads to entropic disorder of the β-cristobalite structure between tetrahedra (0.35 nm < r < 0.60 nm) as suggested by the XPDF and NPDF analyses. This again indicates the tetrahedra tilting is frustrated rather than cooperative.

The tetrahedra connectivity in each orientation is important, as a single unit cell can have four single atom types in the [110] direction, and only one single atom type in the [111] direction. This explains why the PZO [110] orientation is more sensitive to tetrahedra–tetrahedra disorder and has more imperfect agreement between the PZO NPDF model and HRSTEM images. However, our PDF analysis indicates that the expected length scale of order is only <0.35 nm, smaller than the thickness of the HAADF STEM samples. For this reason, it is surprising that any

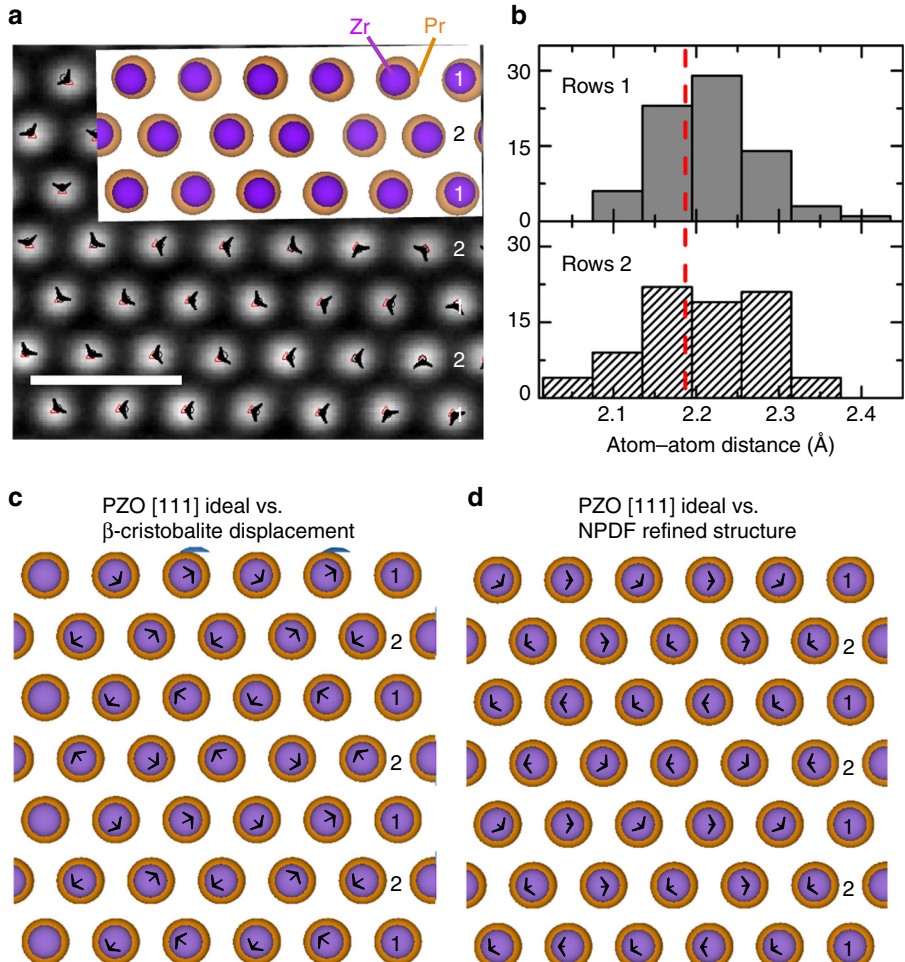

**Fig. 6** $Pr_2Zr_2O_7$ HAADF STEM images and simulations. **a** HAADF STEM images of PZO in the [111] direction, with the $P4_32_12$ structure from NPDF refinements overlaid. Back arrowheads represent a displacement vector from the ideal $Fd\bar{3}m$ structure (black circles) to experimental (red triangles) positions. Scale bar represents a distance of 0.50 nm. **b** Histogram of distance between spots in PZO [111] according to alternating rows 1 and 2, taken from multiple images. The red dashed line represents ideal distance between atomic columns from $Fd\bar{3}m$ Rietveld refinements. **c** Comparison of ideal $Fd\bar{3}m$ structure versus a $P4_32_12$ structure with an exaggerated 0.03 nm Pr β-cristobalite displacement in the [111] direction. Black arrows demonstrate displacement vectors. **d** Comparison of ideal $Fd\bar{3}m$ structure versus a 0.013(2) nm Pr displacement from the NPDF refined $P4_32_12$ structure in the [111] direction. Displaced Pr cations shown in blue. Pr are shown in orange and Zr in purple, with Zr cations reduced by 20% for clarity

**Table 1 Summary of experimental results**

|  | Technique | δ (nm) |
|---|---|---|
| $Pr_2Zr_2O_7$ | SAED | N/A |
|  | SXPD | Pr-0.011(2) |
|  | NPDF* | Pr-0.013(2) |
|  |  | O-0.010(3) |
|  | HRTEM* | Pr-0.013(6) |
|  | $^{91}$Z NQR* | N/A |
| $Yb_2Ti_2O_7$ | SAED | N/A |
|  | XPD | Yb-0.0123(18) |
|  | XPDF* | Yb-0.0098(9) |
|  |  | Ti-0.011(3) |
| $La_2Zr_2O_7$ | SAED[22] | N/A |
|  | XPD | La-0.0094(16) |
|  | NPD | La-0.0079(15) |
|  |  | O-0.0054(13) |

Techniques with an asterisk (*) denotes a methodology which confirms displacements are static with local disorder, rather than dynamic motion. Errors represent statistical uncertainties. *SAED* selected area electron diffraction, *SXPD* synchrotron X-ray powder diffraction, *NPDF* neutron pair distribution function, *XPDF* X-ray pair distribution function, *XPD* laboratory X-ray powder diffraction, and *NPD* neutron powder diffraction.

displacement can be observed in this way, even though it is a statistically real value, and may indicate that the structure is more ordered than anticipated in the [111] direction. To prove the validity of the observed displacement, we further this analysis by comparing the rows in two separate [111] images, taken at different locations on the same sample, with the expected pattern of displacements for the β-cristobalite structure.

A histogram of distances between atoms in each row, taken from multiple images (Fig. 6a and Supplementary Figure 50), is shown in Fig. 6b. Careful observation of comparisons between simulated and PDF refined structures, shown in Fig. 6c, d, respectively, demonstrates that Pr shifts either all left or right in one row (**1**), and alternating left and right in the next row (**2**). Binning these rows separately, using the median and standard deviation (across 153 atomic columns) to create the bin locations and width, Fig. 6b demonstrates one set of rows (**1**) retains a small distribution around a single distance while the other rows (**2**) have two maxima, indicating Pr displaces in alternating directions, as expected for a β-cristobalite displacement in this orientation (as shown in Fig. 6c, d). Additionally, this histogram displays the 0.01 nm deviation from ideal positions, a result well

within error of the cation displacements indicated by PXD, PND, XPDF, and NPDF refinements, and further validates the frustrated tilting of the O′$A_4$ tetrahedra due to the elongation of the O′-$A$ bond.

## Discussion

To understand our results, summarized in Table 1, we need to understand the origin of these displacements. One obvious reason for these displacements is electronic ordering. The β-cristobalite distortion could be caused by a Jahn-Teller effect, where either the central or surrounding atoms could be displaced. Though displacements due to Jahn-Teller effects are typically first-order due to mixing of partially filled $d$ orbitals, $d^0$ transition metals are also known to undergo second-order Jahn-Teller displacements due to mixing of filled $p$ and empty $d$ orbitals[35, 36], and displacements of $B$ in the $BO_6$ octahedra would in-turn displace the surrounding O and $A$ cation. If the displacements observed here are an effect of $d^0$ cations, then only pyrochlores with $d^0$ transition metals would be affected. However, Ti prefers the 32$e$ site (2I2O displacements) while Zr, which has a larger cation radius that farther splits its $p$ and $d$ orbitals[36], prefers the 96$g$/96$h$ site (β-cristobalite displacements). This observation might indicate that second-order Jahn-Teller displacements prefer in-out rather than β-cristobalite type displacements. Knowing the symmetry of the mixing $p$ and $d$ orbitals would help resolve this, as the symmetry of the displacement must match the symmetry of the mixing HOMO and LUMO bands[36].

Another obvious reason for these displacements is inherent strain, well known to exist in the pyrochlore structure[5, 23, 37–40]. The fluorite structure, which resembles the pyrochlore structure when $A$ and $B$ cation sizes are similar and equally mixed, follows body center cubic packing. The pyrochlore structure similarly follows face center cubic packing, but the $B_4$ tetrahedra are smaller than the O′$A_4$ tetrahedra. For large $A$/$B$ ionic radii ratios, the size of the O′$A_4$ tetrahedra are constricted in size by the smaller $B_4$ tetrahedra. Much like β-cristobalite, the tetrahedra distort, allowing longer $A$–O′ bond distances, which relieve the constrained O′$A_4$ tetrahedra, and displace the surrounding O. If this is the case, then β-cristobalite disorder would be present in all pyrochlores with larger $A$/$B$ ionic radii ratios.

Tabira, Withers et al. have observed diffuse scattering for myriad zirconate and titanate pyrochlores[5], indicative of β-cristobalite distortions[27], proposing that tetrahedral distortions occur in non-magnetic $La_2Zr_2O_6O′$ to relieve La–O′ over-bonding[22]. In contrast to previous work, we additionally rule out the effects of impurity elements, site mixing or stuffing, and off-stoichiometry, by analyzing crystals grown as part of series, whose physical properties match those of stoichiometric powders[23, 24], further confirmed by Rietveld and PDF refinements. Given that our work also concludes that the diffuse scattering is due to static, rather than dynamic, displacements, we generalize that *if* these displacements are due to $A$/$B$ cation size mismatch, then static displacements exist in *all* zirconates and titanates at the very least. EXAFS investigations on the ruthenate $Pr_2Ru_2O_7$ likewise found evidence for Pr disorder and deviations in the Pr–O′ bonding[41]. These types of displacements may even explain the structural details for other pyrochlores as well, such as Y disorder in the recently contested $Y_2Mo_2O_7$[11, 25, 29] or the disorder observed in hafnates[42]. If this is the case, then these static displacements, which exist at room temperature, have a significant impact on the properties of these materials.

This structural disorder adds to the number of entropic possibilities at low temperatures. In geometrically and magnetically frustrated compounds, like pyrochlores, this would induce a large array of degenerate magnetically ordered states, creating many

zero-gap or small gap equivalent ground states, and preventing such a material from reaching a theoretical ground state. This does not necessitate that frustration is magnetically driven; rather our results suggest that the frustration is structurally driven with significant impacts on the magnetic ordering, enhancing spin-liquid behavior and at the very least creating spin-glass behavior. Such effects have recently been observed[43, 44], and are important for understanding the underlying physics of the spin-ice state. More thoroughly, a β-cristobalite distortion may not significantly alter bond distances, but bond angles, such as $A$–O–$A$, change drastically, and next-nearest neighbor distances are altered even more dramatically. Hence these small changes in bond distances have extremely large effects on magnetic interactions, explaining why 1–2% changes in stoichiometry have drastic changes in magnetic susceptibility and heat capacity for pyrochlores[23, 45]. Likewise, this also explains why 2% level adjustments in the cation stoichiometry of PZO toward excess Pr leads to a sharper $^{91}$Zr NQR line. As the larger Pr is placed on the smaller Zr site, it allows for the O′$Pr_4$ tetrahedra to expand without distorting, creating more uniform next-nearest neighbor distances. This suggests a design principle of tuning the stoichiometry of these materials to obtain desired properties.

Curiously, ideal stoichiometry leads to optimal magnetic ordering, despite non-uniform magnetic interactions. For instance, any deviation away from nominal stoichiometry in $Yb_2Ti_2O_7$, lowers the temperature of the transition seen in heat capacity data;[24] as placing the smaller Ti on Yb sites (stuffing) requires the O′$Yb_4$ tetrahedra to expand less while placing the larger Yb on the smaller Ti sites (anti-stuffing) allows the O′$Yb_4$ tetrahedra to expand without distorting, both of which normalizes the next-nearest neighbor distances. In both situations, more uniform magnetic interactions should exist, but for both stuffing and anti-stuffing the transition temperature and magnitude of the heat capacity significantly decreases[24]. This indicates that these delicate magnetically ordered states, which already contain magnetic disorder, are easily interrupted by either a lack of magnetic cations on the $A$-framework, or extra magnetic cations on the $B$-framework. Site-mixing would additionally interrupt the delicately magnetically ordered degeneracies, suggesting that ideal magnetic order in pyrochlores cannot exist without non-uniform magnetic interactions.

Signatures of this disorder should be readily observable in spectroscopic measurements, arising as a continuum of non-magnetic, temperature independent, low energy excitations. Resonant inelastic X-ray scattering has observed such a feature for $Eu_2Ir_2O_7$, $Pr_2Ir_2O_7$, and $Sm_2Ir_2O_7$[8, 46], though its appearance is likely complicated by other excitations. This feature unexpectedly extends to higher energy for $Pr_2Ir_2O_7$ than $Eu_2Ir_2O_7$[8], as a result of the larger cationic ratio of Pr/Ir, inducing more disorder for $Pr_2Ir_2O_7$. Similarly, because the local symmetry is lowered, higher energy excitations would also appear – though their intensities would be severely diminished based on the magnitude of the displacements, as is the intensity of the allowed {442} reflection for diffraction data.

Though space group $P4_32_12$ may not accurately describe the disorder between tetrahedra, it is the highest symmetry space group which allows for tetrahedra to rotate in all three directions, and conveniently contains only one symmetry-equivalent $A$ and $B$ cation. Similarly, the disorder due to frustrated tilting of the tetrahedra, prevents the pyrochlore structure from being ferroelectric, despite the local structure deviating from cubic symmetry. It is this same local disorder which prevents these materials from ordering in their ferroelectric ground state[16, 18] that also enhances the dielectric properties. This lattice frustration has been previously proposed, though only in the context of lone-pair active cations with large displacements[21]. Our results

generalize that hypothesis to likely include all pyrochlores, even when they contain minimal defects, which emphasizes that frustrated tetrahedral motion is the defect-free pyrochlore ground state. The inherent geometric frustration of the pyrochlore structure necessitates that the displacements, and hence dipoles, point in different directions, canceling each other out over a single, or several, unit cells. This behavior prevents displacive spontaneous polarization such as for perovskites $BaTiO_3$ or $PbTiO_3$, but enhances dielectric behavior due to small, localized electric dipoles. Given enough disorder to relieve the geometric frustration, i.e. O vacancies and $A/B$ site mixing, long-range order and ferroelectric behavior might be recovered and is likely the origin of ferroelectric behavior in materials such $Cd_2Nd_2O_7$ and $Pb_2Nd_2O_7$[47].

Though the diverse variety of techniques herein, with multiple complementary techniques applied to $Pr_2Zr_2O_7$, $La_2Zr_2O_7$, and $Yb_2Ti_2O_7$, show both indirect and direct evidence for a static displacement in pyrochlores in the absence of any type of defects or electronic effects, further experiments, which look at atomic positions as a function of temperature, would prove even more conclusively that this displacement is static, and not dynamic. However, if these displacements are due to $A/B$ cation size mismatch they exist in all pyrochlores and not only resolve several long standing issues, such as the discrepancy between experimental and calculated pyrochlore lattice parameters[5, 48–51], but also many recently contested issues[11, 25, 42]. Since the $A$-O′ bond distance appears longer, these conclusions indicate that the bond valence sums, which were revised for the bonding coordination in these compounds[52], may need to be updated. Since these displacements are frustrated, they also explain why pyrochlores are excellent dielectrics but almost never ferroelectric; this same structural frustration also explains why pyrochlores are excellent candidates for spin-ice/liquid behavior but also extremely sensitive to defect concentrations.

## Methods

**Synthesis**. Stoichiometric single crystals of LZO, PZO, and YTO were prepared by the optical floating zone method[53]. Growth conditions were individually optimized to produce homogenous single crystals with physical properties indistinguishable from stoichiometric polycrystalline samples. LZO was prepared by grinding stoichiometric amounts of $La_2O_3$ (99.99%, Alfa Aesar) and $ZrO_2$ (99.978%, Alfa Aesar), after they were individually dried overnight at 1000 °C. Large sample masses were used to minimize mass error (~20 g) and final oxygen content was corrected by heating in ambient atmosphere. Ground, mixed precursors were placed into an alumina crucible and heated under ambient atmosphere to 1350 °C for 12 h. Upon further heat treatments at 1500 °C for 40 h, with intermediate grindings and pelletizing, powder X-ray diffraction confirmed phase purity. Polycrystalline feed rods made from the powder were then sintered at 1500 °C in air. Finally, a pure, stoichiometric single crystal (~5 mm diameter, ~30 mm length) was grown from the feed rods in a four mirror image furnace (Crystal System Inc. FZ-T-12000-X-VPO-PC equipped with four 3 kW halogen lamps) using a growth rate of 4 mm/h and a rotation rate of 0.1 Hz. Details of similarly prepared PZO and YTO single crystals are found in refs. [23, 24] respectively, as well as their associated sintered powder equivalents.

**Diffraction**. Laboratory powder X-ray diffraction patterns were collected for ground single crystals of PZO, LZO, and LTO using Cu-Kα radiation ($\lambda_{avg} = 0.15418$ nm) on a Bruker D8 Focus diffractometer with LynxEye detector at 295 K. Synchrotron powder X-ray diffraction was collected for a ground piece of the PZO single crystal, on the high-resolution 11-BM-B diffractometer at the Advanced Photon Source (APS), Argonne National Laboratory (ANL), with an incident wavelength of $\lambda = 0.0413973$ nm at 298 K. $SiO_2$ was added to attenuate absorption and Si was added as an internal standard. To verify phase purity Rietveld analyses were conducted with Topas Academic and GSAS-II[54], then systematic tests were conducted to verify stoichiometry, $A/B$ cation mixing, and alternative Wyckoff positions as discussed in Supplementary Discussion. Structures were visualized using VESTA[55].

Room temperature time of flight neutron diffraction was collected on a ground single crystal of $La_2Zr_2O_7$ (LZO) at the POWGEN powder diffractometer at the Spallation Neutron Source (SNS), Oak Ridge National Laboratory (ORNL). The data were collected with a central wavelength of 0.1066 nm.

Room temperature X-ray pair distribution data (XPDF) was collected on a well-ground piece of the YTO single crystal on the 11-ID-B diffractometer at the APS, ANL, with an incident energy of 58.6 keV. A $CeO_2$ standard was used to estimate the resolution of the instrument. Data was reduced using Fit2D[56] and PDFgetX2[57]. Time of flight neutron pair distribution data (NPDF) was collected on a piece of well ground, sintered, stoichiometric PZO feed rod, on NOMAD diffractometer at the SNS, ORNL[58]. The 3 mm diameter quartz capillary containing the sample was held in the beam using a linear automated sample changer. Data were collected on the sample and an empty quartz capillary at 298 K for 415 min for each specimen. The scattering data were corrected by subtracting background scattering from the instrument and empty capillary, and normalizing against the scattering from a solid vanadium rod. The PDFgui software package was used for PDF refinements[59], including systematic tests to verify stoichiometry, A/B cation mixing, and alternative Wyckoff positions.

**⁹¹Zr NQR**. ⁹¹Zr NQR measurements on a piece of stoichiometric, sintered PZO were conducted with a conventional pulse-echo method at 4.2 K. The frequency swept NQR spectra were obtained by recording the integrated intensity of the spin-echo signal at discrete frequencies. Since the Zr sites have threefold rotational symmetry around the [111] crystallographic axis, the electric field gradient (EFG) around the Zr sites are axially symmetric, $V_{xx} = V_{yy} = -1/2\ V_{zz}$. Then, two NRQ lines appear at $v_Q$ ($= 3eQV_{zz}/2I(2I-1)h$) and $2v_Q$ in a zero external field, where $Q$ and $I$ are the quadrupole and the spin moment of the ⁹¹Zr nucleus, respectively, and $h$ is the Plank constant. These NQR lines correspond to the transition $I_z = \pm 1/2 \leftrightarrow \pm 3/2$ and $I_z = \pm 3/2 \leftrightarrow \pm 5/2$. The latter transition lines were observed at $2v_Q \sim 17$ MHz. Fits for the ⁹¹Zr NQR spectra were conducted using a Lorentzian function given by Supplementary Eq. 1. Fitting range was from 16.9 to 19.4 MHz to ensure comparative fits were conducted.

$$y = w[1] + \frac{w[2]}{(x - w[3]) + w[4]} \qquad (1)$$

**Electron microscopy**. Selected area electron diffraction (SAED) was conducted on ion-milled slices of PZO, LZO, and YTO single crystals, oriented down the [111] direction, using a Phillips CM300 FEG TEM, operating at an accelerating voltage of 297 kV. Off-center images were collected on an Orius CCD camera, while centered images were collected on film (Kodak SO 163). Patterns were collected under a large variety of exposure times (10 s–10 min) to determine the best exposure time to highlight the diffuse scattering, as demonstrated by Supplementary Figures 42, 43. Additional intensity scaling tricks, such as changing image brightness/contrast and inverting the colors, were conducted. A beam stop was not used both because the long exposure times still resulted in beam spilling, and to observe the completeness of the diffuse scattering.

High-angle annular dark field (HAADF) scanning transmission electron microscopy (STEM) imaging was conducted on the PZO and LZO samples using a Nion UltraSTEM 100 with a C5 Nion QO aberration corrector, operated at 100 kV at the Daresbury SuperSTEM Laboratory. The optics of the microscope were adjusted to form a 0.09 nm probe with 60 pA current. The beam convergence semi-angle was 30 mrad, while the detector angular range was calibrated at 85–190 mrad. In order to take advantage in recent developments in high-precision HAADF imaging through (non-)rigid registration techniques[60], series of up to one hundred successively scanned images were systematically collected over a duration of several seconds. Subsequently, each image was realigned by an autocorrelation algorithm (and visual inspection for completeness) to remove time dependent drift before summing the series to form the final image. No filtering was performed on the images. Alternatively, the non-rigid registration algorithm SmartAlign was applied to some of the images to provide the highest precision imaging possible[61]. The automatic peak finding algorithm Ranger[33] was used to extract experimental atomic positions from the STEM data, which uses 2-D Gaussian fit to obtain sub-pixel precision on the location of each column.

EELS was carried out using a Gatan Enfina spectrometer, with a collection semi-angle of 36 mrad. Spectrum images were acquired by rastering the probe serially across a pre-defined area of the sample and collecting both HAADF and EELS signals at each point. Compositional maps were extracted from these datasets by integrating the relevant intensity above the Zr and Pr $M_{4,5}$ EELS edge onsets over a 50 eV energy window, after subtraction of the continuously decaying background using a power law model. Prior to integration, the signal was denoised using principal component analysis[62], taking great care that no artifact was introduced by this process through thorough inspection of the residuals. To remove the effects of multiple scattering, the signal was also deconvoluted using the Fourier-ratio method[63] using a low-loss spectrum image acquired immediately after the core loss dataset – the remarkably low drift and high stability of the instrument making this practical. Finally, the signal intensity for both elements was normalized to unity, such that the color map cannot be interpreted in terms of absolute composition, but nevertheless provide a useful indication on the relative distribution of the chemical elements across the field of view. Numerous chemical maps were acquired, systematically displaying close composition in atomic columns based on the elemental signal. The sample thickness varied from area to area, ranging between 0.45 and 0.70 inelastic mean free paths in the material (which is estimated

to be 67 nm for PZO at 100 kV acceleration voltage. Though the EELS profiles shown in Supplementary Figure 46e follow the expected intensity patterns, the shape of the curves are odd, and several clear pixel outliers are seen in Supplementary Figures 46a, b which, in addition to previous comments, exemplify that this analysis is more qualitative than quantitative. Supplementary Figure 47 shows the same map used in the main text, along with the individual Pr and Zr maps, all processed as above.

To further quantify the amount of site mixing in the PZO sample, we analyzed line profiles of a HAADF STEM [110] image, which contains rows of mixed atomic columns and rows with alternating Pr/Zr atomic columns. Given that powder diffraction and PDF refinements indicate no site-mixing or off-stoichiometry, we assume that site mixing is a negligible contribution in rows with mixed cations columns; instead, effects such as varied sample thickness, sample bending, and multiple scattering dominate intensity fluctuations for mixed rows. Hence, the standard deviation of a mixed rows accounts for only experimental deviations of intensity. Under this assumption, the errors for each atom type on alternating rows should be the same or less than that for mixed rows if no site-mixing or off-stoichiometry is present.

To obtain the intensity maximum for each column a Gaussian was fit to each peak using DigitalMicrograph, and profiles were lined up for better comparison. The average intensity for mixed columns was 22.4(2) $10^6$ a.u. (taken over 60 atomic columns), was 21.92(19) $10^6$ a.u. for Zr columns (taken over 30 atomic columns), and 22.7(2) $10^6$ a.u. for Pr columns (taken over 30 atomic columns). The magnitude of the standard deviation, a 1% deviation from the average intensity for all column types, indicates that each contain similar errors, suggesting merely experimental effects rather than defect chemistry. Given that two standard deviations account for the intensity fluctuation of the mixed column rows, we use two standard deviations as the metric of experimental error for individual rows. Only one atomic column (third Zr column in row 2, shown in Supplementary Figure 48b) is observed to deviate beyond two standard deviations, indicating that 0.83% (1/120) of columns significantly deviate from ideal cation ordering, with no neighboring columns with similar deviations. Using the binomial distribution model suggested by Mostaed et al. to quantify the stuffing defect concentration[32], and adjusting the defect concentration to match with the experimentally observed number of columns whose intensity deviated (1/120, mentioned above), using a experimentally consistent 44 nm thick sample (119 atoms in the [110] orientation), suggests a defect concentration of 0.007%. Similarly, if we assume that this column contains two defect atoms, it would still only require a defect concentration of 0.11%. Finally, if we assume the sample is half as thick, with eight times as many columns with defects, this still only suggests a defect concentration of 0.12%, indicating that the defect concentration is well below the 0.5% limit proposed by powder diffraction and PDF refinements.

To obtain displacement vectors for high-angle annular dark field (HAADF) scanning transmission electron microscopy (STEM) images simulated pattern of cation spots were matched and refined to the experimental images using MATLAB. Simulated patterns were created using $Fd\bar{3}m$ lattice parameters derived from room temperature powder diffraction and image length scale. An example is shown for PZO in the [110] and [111] orientations in Supplementary Figure 47.

Using a least-squares analysis the calculated patterns were translated and rotated, and finally scaled to account for slight lattice mismatch or sample tilting, though scaling never exceeded a 3% difference. The edges of final images were cropped due to an observed "fish-eye" type image distortion, shown in Supplementary Figure 51a, where the middle of the image fits quite well while the outskirts fit poorly. Supplementary Figure 51b provides a comparison of fitting in several quadrants of the same image, which all fit exceptionally well, providing justification for using fits to the center of more zoomed in images. We hypothesize this image distortion is physical due to bending of the sample as it was more drastic for further zoomed out images. Such behavior is expected given the known strain in these materials[5, 23, 37–40], and given the proposed structural strain from A/B cation size mismatch.

Finally, to obtain an average displacement distance between sub-pixel 2D Gaussian refined experimental distances and the least square refined ideal structure, we use the distance formula to obtain the distance between the spots in sub-pixels, convert this distance to nanometer using the image scale, and finally averaged all the calculated displacements, over 45 atomic columns, to obtain 0.013(6) nm.

**Data availability**. The authors declare that the data supporting the conclusions of this study are available within the article and its Supplementary Information file or from the corresponding author upon reasonable request. The crystallographic data (Supplementary Data 1, 2) can be obtained free of charge from The Cambridge Crystallographic Data Centre via www.ccdc.cam.ac.uk/data_request/cif (CCDC-1520136-1520137).

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

## Acknowledgements

The work at IQM was supported by the US Department of Energy, Office of Basic Energy Sciences, Division of Materials Sciences and Engineering under grant DE-FG02-08ER46544. Use of the Advanced Photon Source at Argonne National Laboratory was supported by the US Department of Energy, Office of Science, Office of Basic Energy Sciences, under contract No. DE-AC02-06CH11357. Use of ORNL's Spallation Neutron Source was sponsored by the Scientific User Facilities Division, Office of Basic Energy Sciences, US Department of Energy. The SuperSTEM Laboratory is the UK National Facility for Aberration-Corrected STEM and is supported by the Engineering and Physical Sciences Research Council (EPSRC). This work was also partly supported by the David and Lucile Packard Foundation and the Johns Hopkins University Catalyst Fund. B.A. Trump would also like to acknowledge Lewys Jones and Ally Fry for helpful discussions. Certain commercial equipment, instruments, materials, or software are identified in this paper to foster understanding. Such identification does not imply recommendation or endorsement by the National Institute of Standards and Technology, nor does it imply that the materials or equipment identified are necessarily the best available for the purpose.

## Author contributions

All authors have given approval to the final version of the manuscript. S.M.K., J.-J.W., K.E.A., and B.A.T. prepared samples and collected diffraction data. M.F. helped conduct neutron PDF acquisition. K.J.T.L., Q.M.R. and R.B. were involved with HR-STEM measurements with analysis conducted by K.J.T.L., Q.M.R., and B.A.T.. H.T., M.T., K.K., and S.N. conducted and analyzed NQR data. C.L.B. and T.M.M. initiated and oversaw the project. B.A.T. did structural refinements and wrote the manuscript with input from all authors.

## Additional information

**Competing interests:** The authors declare no competing interests.

