## [Peer Review File · Nature Communications]

Editorial Note: Parts of this Peer Review File have been redacted as indicated as we could not obtain permission to publish the reports of reviewer #2.

Reviewers' comments:

Reviewer #1 (Remarks to the Author):

I would suggest this paper is publishable with major revisions and some re-analysis of the data, recollection of some data might also be required. This paper has attempted to report, clearly, the observation of structural instability present in the pyrochlores $A_2Zr_2O_6O'$ ($A = Pr, La$) and $Yb_2Ti_2O_6O'$, that exists despite ideal stoichiometry, ideal cation-ordering, no lone pair effects, and a lack of magnetic order. Though these materials appear to have good long-range order, local structure probes find displacements, of the order of 0.01 nm, within the pyrochlore framework. Unfortunately, I believe the work, in its current state falls short of this goal. Please see comments in reference to both the manuscript and the supplemental portions.

This paper also claims to be applicable universally as such displacements within the pyrochlore structure adds to the known structural diversity and explains the extreme sensitivity to composition found in quantum spin ices and the lack of ferroelectric behavior in pyrochlores. It is this reviewer's opinion that this statement may be true, but the evidence provided in this manuscript does not prove or disprove this statement. As well, it would be nice to comment briefly on how this increased knowledge could benefit these area? While this finding would be important and can be used for future predicative modelling and explanation, the type of frustration described in this paper has been described in a number of other structures previously, as cited by this paper.

Supp, p4: Though alternative Wyckoff positions, ADE, and diffuse electron scattering hint as disorder, they do not differentiate between dynamic and static displacements. Only the addition of forbidden reflections or extra intensity for allowed reflections can indicate that displacements are long-range ordered and static. Alternatively, local probes, such as pair-distribution function (PDF) analysis, NMR, X-ray absorption fine structure spectroscopy, or high-resolution transmission electron microscopy (HRTEM), can distinguish between static or dynamic distortions. However, their effectiveness is limited by the timescale and resolution of the measurements – as well as the magnitude of the displacement.

This statement is accurate in the reviewer's opinion, however, the data presented in this paper does not appear to uniquely or fully clarify the above statement. Questions arise

about the cause of the diffraction intensities due to the lack of modelling to verify the unusual streaking present in the SAED patterns. Additionally, the overexposure raises concern about collection artifacts. HAADF-STEM imaging is directly limited by the resolution of the microscopy and with statistical measures it has been shown that picometer precision is possible under ideal imaging conditions. There is no evidence in this paper or supplemental supporting that 0.01nm precision was achieved.

Supp, p9: "We hypothesize this image distortion is physical due to bending of the sample as it was more drastic for further zoomed out images. Such behavior is expected given the known strain in these materials, ^{3,20,31-34} and given the proposed structural strain from A/B cation size mismatch. "

It should be possible to determine this? But what type of magnification are we referring to? The images presented are from very local regions, encapsulating a few unit cells. The type of bending and distortion would be expected over much larger areas than the data presented in this paper. Often the bending is non-uniform and follows thickness variations? The fact that a fish eye formed does raise concern as to where the variation in the overlaid structure arises?

P4: Here we provide evidence for this same disorder in the pyrochlores Pr₂Zr₂O₇ (PZO), La₂Zr₂O₇ (LZO), and Yb₂Ti₂O₇ (YTO), due to static A and O displacements (Figure 1c). These displacements mimic isorecticular β-cristobalite, where corner-sharing tetrahedra cooperatively tilt, allowing for an increase in the Si-O bond length and a deviation of the O-Si-O angle from 180° (Figure 1d).¹⁸

From the reference, the SAED for β-cristobalite is much cleaner and is displayed as normal SEAD patterns are, with balanced contrast and little to no oversaturation. Please see statements on Fig.2

P7: Figure 5a shows a corresponding electron energy loss spectroscopy (EELS) spectrum image where red represents Pr, blue represents Zr, and purple represents mixed columns, demonstrating that PZO has ideal the A/B cation mixing as indicated from Rietveld refinement tests.

Fig 5a is an image and color map with no legend or spectra. This in no way indicates or confirms that the PZO has and ideal A/B cation mixing. This is a very specific statement that requires quantitative EELS, ie K-K analysis and zero loss deconvolution. If this

level of analysis was performed, it should be presented in the supplemental.

Currently, no EELS spectra are presented and this makes Fig 5a ambiguous. It is simply a color map. I could find no mention of the thickness of the sample or for understanding the multiple scattering in the sample. This point should be addressed.

P7: Ranger was used to identify experimental HRSTEM atomic columns and ideal atomic positions were calculated from the $Fd\bar{3}m$ structure for comparison.

This technique is not well known, and should be included in the supplemental to understand how precision of 0.01nm is achieved. Explicit description as well as an example measurement would be appreciated. Can you determine an error bar for this level of precision?

P8: The observed displacements, of the order of 0.01 nm, are markedly larger than the precision typically achieved with this workflow (consecutive acquisition followed by rigid or non-rigid registration) with this instrument.²⁵

It is not clear how a value of 0.01nm is achieved, and it is not obvious.

Figure 2: These patterns are considered unusable by electron microscopist's standards.

the reviewer explored references:

Tabira, Y., Withers, R., Thompson, J. & Schmid, S. *J. Solid State Chem.* **142**, 393–399 (1999) G.L.Hua,T.R.Welberry,R.L.Withers,andJ.G.Thompson,*J.Appl. Crystallogr.* 21, 458 (1988).

R.L.Withers,J.G.Thompson,andT.R.Welberry,*Phys.Chem.Miner.* 16, 517 (1989).

R. L. Withers, S. Schmid, and J. G. Thompson, *Prog. Solid State Chem.* 26, 1 (1998).

In 1999 these patterns were collected improperly using film and the current author has repeated this process it appears. A beam stop should be used to block the transmitted beam and reduce oversaturation. The experimental image exhibits an extreme saturation of the central portion of the film and it also appears that there might have been over development of the film as the central bragg reflections are bright, as opposed to the outer reflections which are dark. The reviewer can suggest a “farmer’s reducer”, this can be explored on the internet and will reduce the density of the

overexposed negative. Additionally, this can be an artifact from the scanning, if you have a reflective scanner as opposed to a transmission scanner, this effect may appear.

The first step to preventing this type of oversaturation is to insert a beam stop to block the transmitted beam. In 2017, the best method for doing this is to use a digital camera as opposed to film. As part of this work was performed at SuperSTEM, they have outstanding equipment and digital cameras to acquire patterns with high dynamic range and observe both the primary reflections and weaker intensities.

While this previous work was published 15+yrs ago, there has been no modern revisit to this analysis that I have been able to find. It would be very elegant to show a simple model with modern software that would incorporate the tetrahedral shifting that you are referring to in this paper. There are currently software packages available that will allow input of given structures and fully model elastic and inelastic interactions to produce simulated diffraction patterns. This would shed light on the previous research and help support the previous hypothesis as to where the diffuse streaking arises. More importantly, it will strengthen this papers statements and claims.

From an analytical standpoint, in
R.L.Withers,J.G.Thompson,andT.R.Welberry,Phys.Chem.Miner. 16, 517 (1989).

The SAED patterns are not oversaturated and most are not off zone. Can the authors comment on why it was necessary to collect the patterns under these conditions? Also, can the authors explain why the diffuse streaking is not present uniformly in their SAED patterns.

Reviewer #2 (Remarks to the Author):

[Redacted]

Reviewer #3 (Remarks to the Author):

I read the manuscript by Trump et al. few times. I can now declare that it is the densest manuscript I have ever read. It shows: selected area electron diffraction, neutron scattering, x-ray scattering, nuclear quadrupole resonance, scanning transmission electron microscopy, and electron energy

loss spectroscopy data. These techniques are applied to three different samples but not all techniques are applied to all samples. It is very difficult to follow the manuscript, understand the authors message, and to relate a data set to a conclusion. It doesn't help that the manuscript ends with the statement "These displacements also have significant implications for quantum magnetism and dielectric behavior" without stating what are the implications. Nevertheless, I do believe the manuscript has "significant implication", and with a bit of work on the message of the paper it should be published in Nature communication.

If I understood correctly, what the authors are trying to tell us is that there are three questions they want to address:

- 1) Why spin ice properties are extremely sensitive to composition?
- 2) Why are pyrochlores not Ferroelectric?
- 3) Is magnetism relevant for their distortion?

To address these question one needs to determine the nature of the distortion including space group and whether it is static or dynamic. It is also important to know if it exists in non-magnetic materials.

Question 1 is intimately related to the type of distortions existing in the pyrochlore as reviewed in the second paragraph, but the link is not made in the paper. Question 2 receives satisfactory attention in the introduction in the third paragraph. Question 3 is not explained at all.

I recommend the author to state very clearly the questions, to which question is their finding significant, and keep refereeing to the "big questions" before and after they present each data set so we know why are we supposed to read the next paragraph and what we learned from it. I also recommend the author to mention again in the conclusions what are the "significant implication" of their work.

I have some more detailed comments as well:

- 1) "quantum magnetism" is not an application.
- 2) "Figure 1a" should be "see Figure 1a".
- 3) "due to split peaks in X-ray absorption spectroscopy". X-ray absorption spectroscopy is not the reason for the split, it the method with which it was detected.

- 4) "no lone pair effects" should be "no long pair effects".
- 5) "despite La₂Zr₂O₇ being nonmagnetic." should be "despite La₂Zr₂O₇ being nonmagnetic contrary to previous suggestions [*]".
[*] A. Keren and J. S. Gardner PRL 87, 177201 (2001).
- 6) From the sentences following "Additionally, local structure probes..." one can understand that NQR is not a static probe. Is that what the authors mean?
- 7) "PZO has ideal the A/B cation" should be "PZO has the ideal A/B cation".
- 8) HRSTEM is not defined.
- 9) "There are a variety of reasons for these displacements to occur, such as electronic ordering". This is only one reason. The reader is expecting more.
- 10) "are needed to definitively prove that this displacement is static, and not dynamic". In one case it is clear that the displacements are static. See O. Ofer et al., PRB 82, 092403 (2010).

Reviewer #1

I would suggest this paper is publishable with major revisions and some re-analysis of the data, recollection of some data might also be required. This paper has attempted to report, clearly, the observation of structural instability present in the pyrochlores $A_2Zr_2O_6O'$ ($A = Pr, La$) and $Yb_2Ti_2O_6O'$, that exists despite ideal stoichiometry, ideal cation-ordering, no lone pair effects, and a lack of magnetic order. Though these materials appear to have good long-range order, local structure probes find displacements, of the order of 0.01 nm, within the pyrochlore framework. Unfortunately, I believe the work, in its current state falls short of this goal. Please see comments in reference to both the manuscript and the supplemental portions. This paper also claims to be applicable universally as such displacements within the pyrochlore structure adds to the known structural diversity and explains the extreme sensitivity to composition found in quantum spin ices and the lack of ferroelectric behavior in pyrochlores. It is this reviewer's opinion that this statement may be true, but the evidence provided in this manuscript does not prove or disprove this statement. As well, it would be nice to comment briefly on how this increased knowledge could benefit these area? While this finding would be important and can be used for future predicative modelling and explanation, the type of frustration described in this paper has been described in a number of other structures previously, as cited by this paper.

C1: We have re-analyzed multiple sets of data, and included several new analyses in order to significantly improve the manuscript. Additionally, we have more adequately presented previous works, as well as more clearly explained how our results are novel and distinct from these previous works. The very distinction of our work (the fact that we observed these displacements in the absence of any electronic/chemical drivers) is also what makes the result extremely wide-spread. It is also important that this is not an isolated work, rather our novel results are placed in the context of the work of scores of cited researchers. Lastly, we have added additional discussions explaining how the results impact a large variety of topics and scientific fields. Thankfully we can include these extra discussions in the current manuscript, as the previous submission was transferred from Nature Materials, which has a more constrictive word limit.

Supp, p4: Though alternative Wyckoff positions, ADE, and diffuse electron scattering hint as disorder, they do not differentiate between dynamic and static displacements. Only the addition of forbidden reflections or extra intensity for allowed reflections can indicate that displacements are long-range ordered and static. Alternatively, local probes, such as pair-distribution function (PDF) analysis, NMR, X-ray absorption fine structure spectroscopy, or high-resolution transmission electron microscopy (HRTEM), can distinguish between static or dynamic distortions. However, their effectiveness is limited by the timescale and resolution of the measurements – as well as the magnitude of the displacement.

This statement is accurate in the reviewer's opinion, however, the data presented in this paper does not appear to uniquely or fully clarify the above statement. Questions arise about the cause of the diffraction intensities due to the lack of modelling to verify the unusual streaking present in the SAED patterns. Additionally, the overexposure raises concern about collection artifacts. HAADF-STEM imaging is directly limited by the resolution of the microscopy and with statistical measures it has been show that picometer precision is possible under ideal imaging conditions. There is no evidence in this paper or supplemental supporting that 0.01nm precision was achieved.

C2: See comments C4 and C5 on SAED patterns. While no specific data is presented here to confirm the precision, the work cited in the main text and SI, carried out on the same instrument with almost

identical conditions, demonstrated this level of precision is more than possible. These earlier studies were concerned with methodological development and explored in detail the achievable precision, and refined the workflow necessary to reach them. Here the workflows are directly applied to explore possible displacements in samples of an “un-verified” relaxed atomic structure. The measurement precision can therefore only be inferred from previous work, as estimating it from the investigated sample would contradict the premise of the study.

However, in order to comply, we mention more clearly that the displacements observed from HRSTEM come from the difference between 2-D Gaussian fits of atomic columns and the ideal $Fd-3m$ structure. More importantly the magnitude of displacements are identical to those observed from both powder diffraction and pair-distribution function analysis in this work.

Supp, p9: “We hypothesize this image distortion is physical due to bending of the sample as it was more drastic for further zoomed out images. Such behavior is expected given the known strain in these materials, 3,20,31–34 and given the proposed structural strain from A/B cation size mismatch. “

It should be possible to determine this? But what type of magnification are we referring to? The images presented are from very local regions, encapsulating a few unit cells. The type of bending and distortion would be expected over much larger areas than the data presented in this paper. Often the bending is non-uniform and follows thickness variations? The fact that a fish eye formed does raise concern as to where the variation in the overlaid structure arises?

C3: Added Figure S46 which directly shows the fish eye effect, as well as some additional SI information. The observed image displays the reviewer’s point as different areas appear more deformed than others, despite the fit always being poorer on the outskirts of the image. Additionally, given that the center of the image fits equally well with different overall areas, the validity of our technique is seen to be sound.

P4: Here we provide evidence for this same disorder in the pyrochlores $\text{Pr}_2\text{Zr}_2\text{O}_7$ (PZO), $\text{La}_2\text{Zr}_2\text{O}_7$ (LZO), and $\text{Yb}_2\text{Ti}_2\text{O}_7$ (YTO), due to static A and O displacements (Figure 1c). These displacements mimic isorecticular β -cristobalite, where corner-sharing tetrahedra cooperatively tilt, allowing for an increase in the Si-O bond length and a deviation of the O-Si-O angle from 180° (Figure 1d). 18

From the reference, the SAED for β -cristobalite is much cleaner and is displayed as normal SEAD patterns are, with balanced contrast and little to no oversaturation. Please see statements on Fig.2

C4: We agree that more sophisticated recording equipment would be necessary to carry-out a more quantitative interpretation of the SAED, however, here the patterns are merely shown to highlight the presence of highly unusual streaking (previously observed in many pyrochlores). The many examples we provide, shown in new Figures S42 and S43, systematically reveal the presence of these streaks, regardless of how saturated the central beam is. The SAED data therefore plays its role as a “smoking gun” – while other experimental techniques are used to understand the effect more precisely. This is distinct from previous works, as we use four different, complementary, techniques to confirm that this streaking is from static, not dynamic, displacement in pyrochlores.

P7: Figure 5a shows a corresponding electron energy loss spectroscopy (EELS) spectrum image where red represents Pr, blue represents Zr, and purple represents mixed columns, demonstrating that PZO has ideal the A/B cation mixing as indicated from Rietveld refinement tests.

Fig 5a is an image and color map with no legend or spectra. This in no way indicates or confirms that the PZO has an ideal A/B cation mixing. This is a very specific statement that requires quantitative EELS, ie K-K analysis and zero loss deconvolution. If this level of analysis was performed, it should be presented in the supplemental. Currently, no EELS spectra are presented and this makes Fig 5a ambiguous. It is simply a color map. I could find no mention of the thickness of the sample or for understanding the multiple scattering in the sample. This point should be addressed.

C5: We agree with the reviewer that zero-loss deconvolution is sometimes applied to datasets acquired from relatively thick samples, and is typically recommended for cases where the relative thickness is much larger than one inelastic mean free path (λ). The datasets shown here were systematically acquired in much thinner regions, at most $0.85 t/\lambda$, or in the case of the data shown in Figure 5, $0.65 t/\lambda$. For reference $\text{Pr}_2\text{Zr}_2\text{O}_7$ would have an estimated mean free path at 100 kV of about 67 nm, making the sample approximately 44 nm thick.

Regardless, the effect of deconvolution on the qualitative contrast observed in the EELS maps is often minimal if the maps are generated using relatively narrow-energy integration windows, as was the case here (see details in methods) – as plural scattering will mostly affect the edge shape much beyond the onset. However, we performed low-loss deconvolution, and added in additional EELS maps with profiles in Figures S47 and S48 to display the analysis the reviewer requested, however it is clear from the profiles (several pixels are clearly outliers) that this analysis is more qualitative than quantitative.

To further address the A/B cation mixing we additionally have quantitatively analyzed HRSTEM profile intensities, adding Figure S44. Using a previous statistical model, we also estimate a defect concentration. This defect concentration is compared to values from two other separate techniques, all of which agree. These three techniques together confirm the extremely low defect concentration rather than any individual analysis, as well as several previous works, which show that these crystals have the exact same properties of sintered powders of the same stoichiometry. We also added this discussion to the forefront of the story, changing Figure 5b to averaged intensity profiles, and moved Figure 5b to Figure S45 (PZO [110] HAADF STEM).

P7: Ranger was used to identify experimental HRSTEM atomic columns and ideal atomic positions were calculated from the $Fd\bar{3}m$ structure for comparison. This technique is not well known, and should be included in the supplemental to understand how precision of 0.01nm is achieved.

P8: The observed displacements, of the order of 0.01 nm, are markedly larger than the precision typically achieved with this workflow (consecutive acquisition followed by rigid or non-rigid registration) with this instrument. 25

Explicit description as well as an example measurement would be appreciated. Can you determine an error bar for this level of precision?

It is not clear how a value of 0.01nm is achieved, and it is not obvious.

C6: See comment C2 which addresses this thoroughly. We have additionally added in a value, with standard deviation, that comes directly from the difference between 2-D Gaussian fits of atomic columns and the ideal $Fd\bar{3}m$ structure. We also explain the cited Ranger program in more detail.

Figure 2: These patterns are considered unusable by electron microscopist's standards.

the reviewer explored references:

Tabira, Y., Withers, R., Thompson, J. & Schmid, S. J. Solid State Chem. 142, 393–399 (1999)
G.L.Hua,T.R.Welberry,R.L.Withers,andJ.G.Thompson,J.Appl. Crystallogr. 21, 458 (1988).

R.L.Withers,J.G.Thompson,andT.R.Welberry,Phys.Chem.Miner. 16, 517 (1989).

R. L. Withers, S. Schmid, and J. G. Thompson, Prog. Solid State Chem. 26, 1 (1998).

In 1999 these patterns were collected improperly using film and the current author has repeated this process it appears. A beam stop should be used to block the transmitted beam and reduce oversaturation. The experimental image exhibits an extreme saturation of the central portion of the film and it also appears that there might have been over development of the film as the central bragg reflections are bright, as opposed to the outer reflections which are dark. The reviewer can suggest a “farmer’s reducer”, this can be explored on the internet and will reduce the density of the overexposed negative. Additionally, this can be an artifact from the scanning, if you have a reflective scanner as opposed to a transmission scanner, this effect may appear. The first step to preventing this type of oversaturation is to insert a beam stop to block the transmitted beam. In 2017, the best method for doing this is to use a digital camera as opposed to film. As part of this work was performed at SuperSTEM, they have outstanding equipment and digital cameras to acquire patterns with high dynamic range and observe both the primary reflections and weaker intensities.

C7: Sadly, no such equipment exists at the SuperSTEM facility, as they do not specialize in electron diffraction. Hence the electron diffraction was taken on the exact same samples at a different facility. See comments C4 and C8 for more details on the SAED patterns.

While this previous work was published 15+yrs ago, there has been no modern revisit to this analysis that I have been able to find. It would be very elegant to show a simple model with modern software that would incorporate the tetrahedral shifting that you are referring to in this paper. There are currently software packages available that will allow input of given structures and fully model elastic and inelastic interactions to produce simulated diffraction patterns. This would shed light on the previous research and help support the previous hypothesis as to where the diffuse streaking arises. More importantly, it will strengthen this papers statements and claims.

From an analytical standpoint, in

R.L.Withers,J.G.Thompson,andT.R.Welberry,Phys.Chem.Miner. 16, 517 (1989).

The SAED patterns are not oversaturated and most are not off zone. Can the authors comment on why it was necessary to collect the patterns under these conditions.? Also, can the authors explain why the diffuse streaking is not present uniformly in their SAED patterns.

C8: See comment C4 which addresses the SAED patterns in more detail. In addition, the displacements in previous works are markedly larger and easier to observe, thus we were required to collected SAED patterns with longer exposure times, taken slightly off-axis, both of which are common techniques to observe weak diffuse scattering. The diffuse streaking is uniform, except due to the sample being off-axis.

Additionally, modeling this behavior is not trivial, as it would require specialized software and complex, large, low symmetry, unit cells to appropriately capture the short-range order. Again, the observation of streaking here is placed into the larger context of this streaking always being observed for many pyrochlores, and we use four other, separate, complementary, experimental techniques to justify our conclusions.

Reviewer #2 (Remarks to the Author):

[Redacted]

Reviewer #3 (Remarks to the Author):

I read the manuscript by Trump et al. few times. I can now declare that it is the densest manuscript I have ever read. It shows: selected area electron diffraction, neutron scattering, x-ray scattering, nuclear quadrupole resonance, scanning transmission electron microscopy, and electron energy loss spectroscopy data. These techniques are applied to three different samples but not all techniques are applied to all samples. It is very difficult to follow the manuscript, understand the authors message, and to relate a data set to a conclusion. It doesn't help that the manuscript ends with the statement "These displacements also have significant implications for quantum magnetism and dielectric behavior" without stating what are the implications. Nevertheless, I do believe the manuscript has "significant implication", and with a bit of work on the message of the paper it should be published in Nature communication.

C10: Unfortunately, due to many of these state of the art techniques being highly specialized, we were unable to collect the exact same data on all three. This does not however change the conclusions, as multiple data sets (and previous literature data) on all three compounds all have the exact same conclusion – that static displacements exist despite no chemical/electronic drivers. This is made clearer in the text, and Table 1 has been added to summarize which techniques were applied to which materials.

If I understood correctly, what the authors are trying to tell us is that there are three questions they want to address:

- 1) Why spin ice properties are extremely sensitive to composition?
- 2) Why are pyrochlores not Ferroelectric?
- 3) Is magnetism relevant for their distortion?

To address these question one needs to determine the nature of the distortion including space group and whether it is static or dynamic. It is also important to know if it exists in non-magnetic materials.

Question 1 is intimately related to the type of distortions existing in the pyrochlore as reviewed in the second paragraph, but the link is not made in the paper. Question 2 receives satisfactory attention in the introduction in the third paragraph. Question 3 is not explained at all.

I recommend the author to state very clearly the questions, to which question is their finding significant, and keep refereeing to the “big questions” before and after they present each data set so we know why are we supposed to read the next paragraph and what we learned from it. I also recommend the author to mention again in the conclusions what are the “significant implication” of their work.

C11: We have modified conclusions to better address the above points, and added info in main text throughout to keep the reader’s mind focused on these ideas. We also expanded the conclusion to address these ideas specifically, all of which were previously constricted by the Nature Materials word limit, as the initial manuscript was a cross submission.

Specifically, we provide the exact mechanism (A/B cation size mismatch) which drives the displacement, we describe how off-stoichiometry changes interactions (composition sensitivity), describe how three separate techniques show the displacements are frustrated (and hence not ferroelectric), and how this effects magnetism (rather than magnetism effecting it).

I have some more detailed comments as well:

- 1) “quantum magnetism” is not an application. Fixed text.
- 2) “Figure 1a” should be “see Figure 1a”. Fixed text.
- 3) “due to split peaks in X-ray absorption spectroscopy”. X-ray absorption spectroscopy is not the reason for the split, it the method with which it was detected. Fixed text.
- 4) “no lone pair effects” should be “no lone pair effects”. Should be lone pair
- 5) “despite La₂Zr₂O₇ being nonmagnetic.” should be “despite La₂Zr₂O₇ being nonmagnetic contrary to previous suggestions [*]”.
[*] A. Keren and J. S. Gardner PRL 87, 177201 (2001). This paper only mentions Y₂Mo₂O₇. Nothing to do with La₂Zr₂O₇, which from electron counting must be nonmagnetic (unless impurities are present).
- 6) From the sentences following “Additionally, local structure probes...” on can understand that NQR is not a static probe. Is that what the authors mean? Added text to help reviewer better understand concept.
- 7) “PZO has ideal the A/B cation” should be “PZO has the ideal A/B cation”. Fixed text.
- 8) HRSTEM is not defined. Fixed text.
- 9) “There are a variety of reasons for these displacements to occur, such as electronic ordering”. This is only one reason. The reader is expecting more. Fixed text.
- 10) “are needed to definitively prove that this displacement is static, and not dynamic”. In one case it is clear that the displacements are static. See O. Ofer et al., PRB 82, 092403 (2010). Added reference.

Reviewers' comments:

Reviewer #1 (Remarks to the Author):

Thank you for reworking this document. I think it is a nice compilation of how various characterization techniques can be used to help elucidate information on a subject matter. Getting to the point.

1. The SAED patterns are still in a very undesirable form for publication and presentation. As suggested before, a beamstop is required for publication, in my opinion. I know I stated this before, but for what you are trying to collect, a modern CETA, OneView, or any direct electron detection camera would help produce a better SAED than what we have here. While the researchers attempt to collect the various levels of reflections by using longer exposure times, it should be readily apparent to the researchers that the film becomes severely overexposed! This is the point of the beam stop, the transmitted beam is so intense it saturates the entire film over the long exposure, so the hope is to see weak intensities that are washed out by spurious electrons from the intense transmitted beam.

Additionally, presenting raw film SAED patterns is not acceptable, as a first step they should be cropped, they need beam direction labelling and zone axis labelling. The SAEDs need very obvious labelling to guide the reader, most scientists do not know how to interpret SAED intuitively, the lay person is even more ignorant. Where and what is the diffuse streaking in the pattern. This applies to both the on-axis and off-axis SAED.

2. As best I can tell from this document, the researchers attempted to collect and apply the non-rigid registration appropriately, as they state, this technique is not well known, but it has been shown to be accurate, mathematically, assuming no STEM imaging artifacts. The code and programs used for this paper used rapid acquisition of the images using the same scan rotations. A more robust method of this collection technique is to apply scan rotations, this helps alleviate any systematic distortions from the raster pattern, among other things. Understanding the current publications in this area and what is available, this reviewer believes you have accurately applied the technique to your data, but am not convinced that this level of precision achieved. In part due to the channeling and convergence effects between the beam and sample. In a very simplistic view, you state you have ~ 144 atoms in thickness, is the belief that every atom in the viewing direction has shifted, or is it that every atom in the projected view could move in non uniform coordination? If that is the case, then the resultant STEM image is not directly interpretable.

3. Image simulation of this structure should be straightforward and quick. If picometer precision is to be claimed, then images simulation with a perfect structure vs and image with static displacements should be provided for image confirmation.

4. The EELS data is excellent. I think using the term stoichiometric is inappropriate, you try to clarify that this is qualitative, but instead I would take away as many of the quantitative adjectives used in this section and simply state that, there appears to be close composition in given atomic columns based on the elemental signal. It is this reviewer's opinion that clear variation in Pr intensity is observable and that it does not follow the expected periodic pattern of the lattice. From a qualitative standpoint, this may fit your argument, but as one expect very little sample change or beam change over this small area, there is certainly variation in the Pr intensity that would lead a reader to think there is variation in the Pr composition.

Also, to be clear, this thickness of your sample is not important for EELS, it is the thickness of your sample relative to the beam energy that is important. This is why we present EELS measurements in (t/λ) . So, if you were collecting the EELS at 300kV, a 47nm thick sample would be good

for quantitative analysis. However, at 100kV, a 47nm thick sample is almost 2x too thick for quantitative analysis. This is observed in the Zr peak presented in S47.(d), there is a large plasmon that appears after the Zr onset. This large hump is due to thickness. There are also other issues with quantification, but since it is qualitative, this is enough.

Reviewer #2 (Remarks to the Author):

[Redacted]

Reviewer #4 (Remarks to the Author):

The authors have conducted a very thorough investigation of two model pyrochlores (PZO and LZO), but of which are claimed to have minimal defects. The number of experiments and corresponding analysis are admirable. The analysis of the STEM data, however, is somewhat confusing in places and the interpretation is not completely justified. There remain a number of major issues:

1. Line 73: From the results, it is claimed that based on the results the static distortions are "remarkably common" based. This is a fairly over dramatic statement generalizing the observed results from two model systems across pyrochlores.
2. The addition of Figures S2-S28 to the supplement seem overkill, largely because the information is presented in a way that does doesn't particularly help with following the results. Perhaps plots can be summarized equally well in some sort of table or reduced number of figures. A separate supplementary archive containing all the plots for those interested in the specifics may be warranted.
3. Line 105: It is stated that "The broad diffuse scattering bands are Kikuchi bands, which exemplify the excellent crystallinity of these materials." Kikuchi bands do not "exemplify" excellent crystallinity. This is a strange statement without significant meaning. Yes, the selected area is a single crystal, but beyond that, the bands do not really say anything about the crystalline quality here. Also, as pointed to by other reviewers, a beam stop really should have been used here.
4. The authors seem to include all variations of acquisition times used for the film negatives. There is no additional information by including so many and just obfuscates their point. Using up to 60s seems entirely sufficient.
5. Lines (115-116): It is noted that "If the displacements are static, it implies frustrated local disorder, rather than cooperative, tetrahedral tilting, which would perturb long-range magnetic or ferroelectric order." Why is this implied? Diffraction simulations of the tetrahedral tilting would be useful here, if not essential, to help the reader follow the precise geometry they are discussing.
6. Lines (156- 160): The statement "If the displacements are purely dynamic, then the average position should remain the same even if the dynamic motion is anisotropic. In contrast to the timescales of neutron and X-ray measurements due to atomic interactions ($\approx 10^{-12}$ - 10^{-15} s), HAADF STEM is a long timescale measurement, of the order of seconds, much longer than the timescale expected for dynamic motion (at most $\approx 10^{-6}$ - 10^{-12} s)." is likely incorrect. First, the interaction of each electrons with the sample is on the same order as the X-ray interactions. Further, each X-ray measurement in this paper was likely acquired for more time than $1e-12$ to $1e-15$ s and thus averages over a longer time scale just like TEM. Were they using a strobed X-ray probe? The diffraction acquisition details were not provided in the supplement or the manuscript,

but the statement is almost certainly incorrect.

7. Lines (163-164): "The structure appears regular and no noticeable defects (including anti-site phase boundaries)". What is this conclusion based on? Conventional BF/DF imaging (or at least BF STEM with appropriate imaging conditions) would likely be necessary to find these boundaries. No details are provided on this point, so it is difficult to determine how this conclusion was arrived at. Images without defects should be included along with the acquisition parameters (in particular the selected g for two-beam imaging).

8. Line 173: "sit" should read "site"

9. Lines 184 - 188: "Analyzing the individual row intensities, it is seen that only 0.8 % atomic columns deviate from ideal cation ordering. Using the proposed binomial distribution model, even assuming this column contains more than one defect, only suggests a defect concentration of 0.05 %, well below the 0.5 % limit as indicated by powder diffraction and PDF refinements (see SI for details), further indicating that the crystals are nearly defect free." How did the authors arrive at this conclusion? Images simulations are necessary to justify these conclusions, but are not included.

10. I find Figure 5 exceedingly unclear. The overlapping colored circles are not particularly clear. I would suggest making them different sizes. Further, the meaning of the black arrows and corresponding red open circles are not obvious.

11. Line 199: While the measured displacements happen to agree with experiment in the case of PZO, how many images were used to determine this number? If from a single image, then this is not a justifiable conclusion and may have just been a lucky measurement, particularly because the other zone axis did not provide as good agreement. Furthermore, due to overlap of the static displacements down the atom columns, I would not expect good agreement with the X-ray measurements unless all the atoms were displaced by the same amount in the same direction, which would be different than static disorder.

12. Line 249, 251 : The authors state that it is "decisively" determined that diffuse scattering is due to static rather than dynamic displacement. There may still be a mixture and it is not clear based on the data presented for such a strong conclusion. The authors go on to conclude that the observed disorder is due to static displacements "in all zirconates and titanates at the very least". This statement is a seems a fairly strong over generalization from the two samples investigated.

13. The authors seem to go back and forth claiming whether there are pyrochlore ferroelectrics. For example, on line 321, it is claimed that they are "almost never ferroelectric" whereas the abstract claims that this manuscript resolves why pyrochlores lack ferroelectricity.

14. I disagree with the previous reviewer regarding the prescriptive grammarian use of "myriad". The use of "myriad" in the manuscript is now a bit strange from a readability standpoint. I would suggest returning to using "a myriad of" (both are valid from a quick browse through the dictionary, and "a myriad of" has become more common.

Reviewer #1 (Remarks to the Author):

Thank you for reworking this document. I think it is a nice compilation of how various characterization techniques can be used to help elucidate information on a subject matter. Getting to the point.

1. The SAED patterns are still in a very undesirable form for publication and presentation. As suggested before, a beamstop is required for publication, in my opinion. I know I stated this before, but for what you are trying to collect, a modern CETA, OneView, or any direct electron detection camera would help produce a better SAED than what we have here. While the researchers attempt to collect the various levels of reflections by using longer exposure times, it should be readily apparent to the researchers that the film becomes severely overexposed! This is the point of the beam stop, the transmitted beam is so intense it saturates the entire film over the long exposure, so the hope is to see weak intensities that are washed out by spurious electrons from the intense transmitted beam.

Additionally, presenting raw film SAED patterns is not acceptable, as a first step they should be cropped, they need beam direction labelling and zone axis labelling. The SAEDs need very obvious labelling to guide the reader, most scientists do not know how to interpret SAED intuitively, the lay person is even more ignorant. Where and what is the diffuse streaking in the pattern. This applies to both the on-axis and off-axis SAED.

C1: Attempts were made to collect SAED patterns using a beamstop with an Orius CCD camera. Each of these attempts led to diffraction spots with strong enough intensity to harm the detector before we could observe any of the extremely weak diffuse scattering. There was also significant spill-over even with the beamstop at these intensity levels. Additionally, we are unable to collect SAED data on film on that instrument anymore. The best we achieved is to take images with the center beam outside of the field of view, as shown below.

The inclusion of SAED has one purpose in the manuscript. To show that diffuse scattering exists in these materials, to make the direct comparison to Tabira and Withers' works. We agree that the current patterns are over exposed, though it is challenging (as most of their pyrochlore works have shown) to observe the diffuse scattering without over exposing. For this reason, we chose the patterns which are least over exposed to include in the main text, which we believe are publishable. We also believe that it is important to give the reader access to the over exposed images, which we explicitly claim to be over exposed, so they have even more clear evidence that the diffuse scattering exists, and hence are continuing to include them in the SI.

Lastly, we are thankful for the reviewer to remind us to index the SAED patterns, as we caught a critical mistake. The diffuse scattering occurs *perpendicular to* the $\langle 110 \rangle$ directions, not *along* the $\langle 110 \rangle$ directions, in agreement with previous work. Our indexed patterns now properly demonstrate that the diffuse scattering occurs in the $\langle 211 \rangle$ directions, perpendicular to the $\langle 110 \rangle$ directions where disorder occurs.

$\text{Yb}_2\text{Ti}_2\text{O}_7$ [111] SAED, off-center.

$\text{Pr}_2\text{Zr}_2\text{O}_7$ [111] SAED, off-center.

2. As best I can tell from this document, the researchers attempted to collect and apply the non-rigid registration appropriately, as they state, this technique is not well known, but it has been shown to be accurate, mathematically, assuming no STEM imaging artifacts. The code and programs used for this paper used rapid acquisition of the images using the same scan rotations. A more robust method of this collection technique is to apply scan rotations, this helps alleviate any systematic distortions from the raster pattern, among other things. Understanding the current publications in this area and what is available, this reviewer believes you have accurately applied the technique to your data, but am not convinced that this level of precision achieved. In part due to the channeling and convergence effects between the beam and sample. In a very simplistic view, you state you have ~144 atoms in thickness, is the belief that every atom in the viewing direction has shifted, or is it that every atom in the projected view could move in non uniform coordination? If that is the case, then the resultant STEM image is not directly interpretable.

C2: The reviewers make a good point here, in that we claim the displacements are disordered, hence we expect disorder in the STEM images, so it is challenging to accept that we can also observe the static displacements. We agree with this assessment, and would counter that due to these reasons, the HAADF STEM image is not the “smoking gun” for the static displacements, rather it is the histogram in Figure 6d (old Figure 5d). To help with this argument, we have now included an analysis on a separate, larger [111] image, and added that same information into the histogram, meaning that the analysis now includes a comparison between 153 different columns, rather than 45. Even with three times more data, the same trend holds true – one column has short/long distances, and one column has a smaller distribution around a single distance. The expected trend is also more understandable with the added Figures 6c and 6d. Lastly, we added in a discussion in the main text to bring these details to the attention of the reader, as the reviewers do make an important point here.

In addition, the STEM image in Figure 5 was taken at $0.65 \text{ t}/\lambda$, making the sample approximately 44 nm thick. In the [111] direction this comes out to 24 of a single atom type per atomic column (compared as many as 119 in the [110] direction). First off, this alone explains why agreement is poorer in the [110] direction, as more disorder is observable. Secondly, it expected that a column of 24 atom types, or rather corners of 24 separate tetrahedra, is a longer distance than the size of expected order, especially given the PDF results which indicate that disorder could exist between as few as 3-4 of a single atom type.

Despite that, we still end up with an average displacement which is larger than the standard deviation (and somehow is in excellent agreement with a completely different experiment). This suggests that the reported displacement is statistically valid, regardless of the expected disorder. This may be luck, or it may be due to higher ordering in this direction than expected, but either way the reported displacement is a real value. Even averaging between the two images leads to a value of $0.12(8) \text{ \AA}$ which is still statistically a real value.

3. Image simulation of this structure should be straightforward and quick. If picometer precision is to be claimed, then images simulation with a perfect structure vs and image with static displacements should be provided for image confirmation.

C3. This is explicitly what Figure 6a (used to be Figure 5c) is meant to demonstrate, as explained in the text and the caption. The black circles represent the ideal structure, the red triangles represent the experimental positions, and the arrows a vector displacement between them. However, we understand due to the incredibly small size of this displacement it is challenging to observe in this way. As such, we

have split Figure 5 into two figures, one to demonstrate the overall uniformity of the structure, and a second to show the structural displacements. In Figure 6 we have added a comparison of the ideal and displaced structure, with the displacements enhanced for clarity.

4. The EELS data is excellent. I think using the term stoichiometric is inappropriate, you try to clarify that this is qualitative, but instead I would take away as many of the quantitative adjectives used in this section and simply state that, there appears to be close composition in given atomic columns based on the elemental signal. It is this reviewer's opinion that clear variation in Pr intensity is observable and that it does not follow the expected periodic pattern of the lattice. From a qualitative standpoint, this may fit your argument, but as one expect very little sample change or beam change over this small area, there is certainly variation in the Pr intensity that would lead a reader to think there is variation in the Pr composition.

Also, to be clear, this thickness of your sample is not important for EELS, it is the thickness of your sample relative to the beam energy that is important. This is why we present EELS measurements in (t/λ) . So, if you were collecting the EELS at 300kV, a 47nm thick sample would be good for quantitative analysis. However, at 100kV, a 47nm thick sample is almost 2x too thick for quantitative analysis. This is observed in the Zr peak presented in S47.(d), there is a large plasmon that appears after the Zr onset. This large hump is due to thickness. There are also other issues with quantification, but since it is qualitative, this is enough.

C4: Changed wording in the EELS section in the main text to help clarify that it is not exact agreement with stoichiometric compound. Also modified some wording in the SI, to help the reader to conclude that Figure S47d is only qualitative.

Reviewer #2 (Remarks to the Author):

[Redacted]

Reviewer #4 (Remarks to the Author):

The authors have conducted a very thorough investigation of two model pyrochlores (PZO and LZO), but of which are claimed to have minimal defects. The number of experiments and corresponding analysis are admirable. The analysis of the STEM data, however, is somewhat confusing in places and the interpretation is not completely justified. There remain a number of major issues:

1. Line 73: From the results, it is claimed that based on the results the static distortions are "remarkably common" based. This is a fairly over dramatic statement generalizing the observed results from two model systems across pyrochlores.

C5: Removed the word "remarkably". Though we only investigated three materials, we put these results in the context of many other works which noticed similar behavior. At least seven other pyrochlores, where researchers conducted the necessary experiments, have proposed some sort of static local displacements (with direct or indirect observations, and has been proposed for more than 25 others without experimental evidence). It is the observations from all of these works that we generalize this behavior is more common than currently thought.

2. The addition of Figures S2-S28 to the supplement seem overkill, largely because the information is presented in a way that doesn't particularly help with following the results. Perhaps plots can be summarized equally well in some sort of table or reduced number of figures. A separate supplementary archive containing all the plots for those interested in the specifics may be warranted.

C6: The overall trend for each plot is an important observation, as the plots themselves indicate if a certain type of behavior occurs, i.e. whether or not a minimum exists. These results lay out the groundwork for the rest of the manuscript, as they demonstrate that the materials have ideal stoichiometry, and hints at static displacements. We could summarize the plots in a table, but then the reader is left without any evidence of our claims. As such we simply summarize the results in the main text and Table 1 summarizes the most relevant information. At the editor's request we would be willing to split the SI into two documents, one for diffraction information, and a second for microscopy.

3. Line 105: It is stated that "The broad diffuse scattering bands are Kikuchi bands, which exemplify the excellent crystallinity of these materials." Kikuchi bands do not "exemplify" excellent crystallinity. This is a strange statement without significant meaning. Yes, the selected area is a single crystal, but beyond that, the bands do not really say anything about the crystalline quality here. Also, as pointed to by other reviewers, a beam stop really should have been used here.

C7: Sentence changed to: "Broad, diffuse, Kikuchi bands are also observed." You are right that this is simply consistent with the sample (and selected area) being a single crystal. We agree that this statement has no significant meaning here, as the point was to clarify that these are not the features the reader should focus on. See comment C1 for the other points.

4. The authors seem to include all variations of acquisition times used for the film negatives. There is no additional information by including so many and just obfuscates their point. Using up to 60s seems entirely sufficient.

C8: Removed high acquisition time SAED films. We agree that three exposures are significant to show the extreme exposure times which was required to clearly observe the diffuse scattering.

5. Lines (115-116): It is noted that "If the displacements are static, it implies frustrated local disorder, rather than cooperative, tetrahedral tilting, which would perturb long-range magnetic or ferroelectric order." Why is this implied? Diffraction simulations of the tetrahedral tilting would be useful here, if not essential, to help the reader follow the precise geometry they are discussing.

C9: Changed sentence to "If the displacements are static, it implies local disorder, suggesting frustrated, rather than cooperative, tetrahedral tilting,". This is to clarify local disorder must exist if displacements are static, but maybe caused by frustrated tetrahedra tilting.

More thoroughly, if the displacements are static and ordered (cooperative tetrahedral tilting), then symmetry is lowered and no diffuse scattering would be observed. Only if the displacements are static and disordered, or if there is dynamic motion, would the structure appear higher symmetry and have diffuse scattering present.

We observe diffuse scattering and no lowering of symmetry, hence if the displacements are static, they must be disordered over longer ranges. Each Figure S2-S26 compares simulated structures with the experimental data, with alternative Wykoff positions representing different types of motion, with the

crystallographic details mentioned in the text and more thoroughly explained in the SI, including effects on the diffraction pattern. Lastly, Figure 1c and 1d demonstrate the tetrahedral motion.

6. Lines (156- 160): The statement “If the displacements are purely dynamic, then the average position should remain the same even if the dynamic motion is anisotropic. In contrast to the timescales of neutron and X-ray measurements due to atomic interactions ($\approx 10^{-12}$ - 10^{-15} s), HAADF STEM is a long timescale measurement, of the order of seconds, much longer than the timescale expected for dynamic motion (at most $\approx 10^{-6}$ - 10^{-12} s).” is likely incorrect. First, the interaction of each electrons with the sample is on the same order as the X-ray interactions. Further, each X-ray measurement in this paper was likely acquired for more time than 10^{-12} to 10^{-15} s and thus averages over a longer time scale just like TEM. Were they using a strobed X-ray probe? The diffraction acquisition details were not provided in the supplement or the manuscript, but the statement is almost certainly incorrect.

C10: Took out “of neutron and X-ray measurements” as it obfuscates the point that HAADF STEM is an ideal way to verify if motion is dynamic or static.

We agree that both measurements are averaged over time, the difference being that diffraction data is averaged over the sample while STEM data is averaged over an atomic column. This means that can observe a static displacement only when it is ordered over the right length scale, whereas it would still look like dynamic disorder in standard diffraction data.

The diffraction (X-ray, neutron, and PDF) details are included in the Diffraction section of the Methods in the SI.

7. Lines (163-164): “The structure appears regular and no noticeable defects (including anti-site phase boundaries)”. What is this conclusion based on? Conventional BF/DF imaging (or at least BF STEM with appropriate imaging conditions) would likely be necessary to find these boundaries. No details are provided on this point, so it is difficult to determine how this conclusion was arrived at. Images without defects should be included along with the acquisition parameters (in particular the selected g for two-beam imaging).

C11: Added new wording to clarify that we make the statement because there “appeared to be no noticeable defects” during microscopy data collection. During the entire collection time no clear defects were noticed in the atomic structure.

We could include as many as a dozen more [110]/[111] images of different areas on the same sample(s) into the SI, though we feel this unnecessary for this statement. Instead, we added two new zoomed out images to the SI, Figures S44 and S45, orientated in the [110] and [111] directions respectively. We additionally looked at line profiles on both images, and both follow the intensity distributions expected from the absence of anti-site phase boundaries. In fact, in the [110] image you can visually see that there are two rows, one with uniform intensity, and a second with alternating intensity.

Additionally, no data was collected using two beam imaging and all HAADF STEM images were taken using the same settings, only with different magnification.

8. Line 173: “sit” should read “site”

C12: Fixed typo.

9. Lines 184 - 188: "Analyzing the individual row intensities, it is seen that only 0.8 % atomic columns deviate from ideal cation ordering. Using the proposed binomial distribution model, even assuming this column contains more than one defect, only suggests a defect concentration of 0.05 %, well below the 0.5 % limit as indicated by powder diffraction and PDF refinements (see SI for details), further indicating that the crystals are nearly defect free." How did the authors arrive at this conclusion? Images simulations are necessary to justify these conclusions, but are not included.

C13: We added some more details in the main text and SI. For a more thorough explanation:

We began this analysis by looking at the intensity profiles of the HAADF STEM on the [110] image. Out of 120 columns, we observed one column which significantly deviated from two standard deviations (chosen due to the variation in mixed rows), hence 0.83% (1/120) columns deviated. Our sample was 0.65 t/lambda, with Pr₂Zr₂O₇ having and an estimated mean free path at 100 kV of about 67 nm, making the sample approximately 44 nm thick. For this orientation, that means each atomic column contains 119 atoms.

Then, using the binomial distribution equation in Mathematica, we adjusted the defect concentration until the fraction of atomic columns was adjusted to the experimentally observed value of 0.83% (1/120, one out of 120 columns with a defect), leading to a defect concentration of 0.007 %. More specifically:

$$P\left(\frac{X}{Y}\right) = \frac{X!}{Y!(X-Y)!} (1 - p_{xy})^{X-Y} p_{xy}^Y$$

Where X is the amount of atoms in a column (119), Y is the amount of defect atoms per column (1,2,3,...), p_{xy} is the probability of finding an atom of type Y in a column of X atoms (or the defect concentration), and $P\left(\frac{X}{Y}\right)$ is the fraction of atomic columns for each Y value (which is experimentally seen to be 1/120 or 0.83 % for $Y = 1$, in this case). We then adjust p_{xy} until $P\left(\frac{X}{Y}\right)$ results in a value of 0.83 % for $Y = 1$, leading to a defect concentration of 0.007 %.

Additionally, we had noticed that as many as 7 columns slightly deviated from one standard deviation, with one greatly deviated (as mentioned above). Meaning a total of 5.83% columns (7/120) deviating with 1 defect in those columns (meaning $P\left(\frac{X}{Y}\right) = 5.83\%$ when $Y = 1$). The defect concentration was then adjusted to model this experimentally observed value, which additionally allows for a column to contain two defect atoms, leading to a defect concentration of 0.05%.

In addition to the previous work, we calculated a defect concentration for when a column contains two defects at the 0.83 % level (1/120) (meaning $P\left(\frac{X}{Y}\right) = 0.83\%$ when $Y = 2$), additionally modeling more columns with one defect than we experimentally observed. This led to a defect concentration of 0.11 %, which is technically more correct with the current text.

See the plot below, which shows the distribution for 0.007%, 0.05%, 0.11% defects. Given the lengthy SI, and that this is merely using the above binomial distribution equation, we did not think it appropriate to include in the SI, though can include it at the reviewer's request.

10. I find Figure 5 exceedingly unclear. The overlapping colored circles are not particularly clear. I would suggest making them different sizes. Further, the meaning of the black arrows and corresponding red open circles are not obvious.

C14: Zr cations have been reduced in size by ~20% for clarity. Due to the incredibly small nature of this displacement it is challenging to present it in a way which is clear. As such we have split Figure 5 into two figures, with the new Figure 6 showing direct comparisons of an ideal structure with both refined and exaggerated structures. This should make the black arrows/red triangles/black circles, explained in the caption, much clearer to the reader.

11. Line 199: While the measured displacements happen to agree with experiment in the case of PZO, how many images were used to determine this number? If from a single image, then this is not a justifiable conclusion and may have just been a lucky measurement, particularly because the other zone axis did not provide as good agreement. Furthermore, due to overlap of the static displacements down the atom columns, I would not expect good agreement with the X-ray measurements unless all the atoms were displaced by the same amount in the same direction, which would be different than static disorder.

C15: See comment C2, as we have added statistics from a second image to the histogram. Normal diffraction indeed cannot observe the displacements like HAADF STEM, however PDF measurements can observe the local structure and appear to match well.

12. Line 249, 251 : The authors state that it is "decisively" determined that diffuse scattering is due to static rather than dynamic displacement. There may still be a mixture and it is not clear based on the data presented for such a strong conclusion. The authors go on to conclude that the observed disorder is due to static displacements "in all zirconates and titanates at the very least". This statement is a seems a fairly strong over generalization from the two samples investigated.

C16: Removed the word decisively. All six separate analyses do conclude that static displacements are preferred over dynamic motion, though we agree that not all are conclusive, and the disordered nature of the displacements makes it difficult to generalize.

Also reworked the sentence to clarify that the disorder “is due to static displacements in all zirconates titanates”, *only* “if these displacements are due to A/B cation size mismatch”. Cation size mismatch drives the structural frustration, and if a pyrochlore has cation size mismatch, it should contain static displacements – but this does not mean that they could be overshadowed by dynamic motion in some compounds.

It is also worth noting that this is a generalization reached by also considering Tabira and Wither’s work on titanate and zirconate pyrochlores, noting we see the same diffuse scattering that they observed on 3 titanate and 3 zirconate pyrochlores, and proposed it in 20 other pyrochlores.

13. The authors seem to go back and forth claiming whether there are pyrochlore ferroelectrics. For example, on line 321, it is claimed that they are “almost never ferroelectric” whereas the abstract claims that this manuscript resolves why pyrochlores lack ferroelectricity.

C17: This is also a bit confusing in the literature about pyrochlores as well. Pyrochlores have large dielectric constants, and thus should be electronically polarizable, yet they are surprisingly almost never ferroelectric, as explained in the introduction. However, there are a few that exist, though they notably contain significant amounts of defects. Hence pyrochlores can be ferroelectric, but there are surprisingly few of them.

This is why “almost never ferroelectric” and “why they lack ferroelectricity” are consistent. Our observation of the structural frustration in these materials explains why so few ferroelectric pyrochlores exist. The only place in the manuscript where we claim pyrochlores are ferroelectric is the introduction, where it is true that some pyrochlores can be utilized for ferroelectric applications.

The title additionally makes it a bit unclear, hence changed to:

Universal Geometric Frustration in Pyrochlores Driving Quantum Magnetism and (Lack of) Ferroelectricity

14. I disagree with the previous reviewer regarding the prescriptive grammarian use of “myriad”. The use of “myriad” in the manuscript is now a bit strange from a readability standpoint. I would suggest returning to using “a myriad of” (both are valid from a quick browse through the dictionary, and “a myriad of” has become more common.

C18: The word myriad has been replaced in the abstract.

REVIEWERS' COMMENTS:

Reviewer #1 (Remarks to the Author):

Thank you for reviewing the previous comments carefully, you have made nice improvements/corrections. Please see below for comments and suggestion on two of the remaining figures.

Figure 2: The off-center CCD images are beautiful and for the first time I can see the streaking that you are referring too. It would be my opinion that you use these images in the main text and leave the SAED images for the supplemental. Perhaps show the shortest exposure time SAED if you want to show the entire DP, but I think the CCD images show the streaking quite well. It might also be worth adding an additional arrow to help the reader locate it. As well, when thinking about publication, you might want to increase the contrast on the image and up the brightness so that the streaking is more apparent. You can also add indices so that the reader is aware of what reflection they are seeing in the off-axis image.

Figure 6: This is an improvement. It is not easy to glean information from the figure in the current format, however, it is sufficient, I would draw your attention to modern work using DPC imaging and Lorentz TEM/ STEM imaging. This same type of work is showing local changes in magnetic field, but it produces similar data to what you are trying to present in this figure. I highly recommend trying to color map your data to better get your point across. and provide a more elegant appearance. Please see the work presented for Fig 2,3,&4 in this paper by Shibata. DOI: 10.1038/ncomms15631, many other examples exist in various publications, please explore.

For the record, I still have fundamental differences on the ability to use STEM imaging to make the statements you are claiming with regard to the small atomic position displacements, but moving past that, your figures support the text. I would encourage you to think one last time about how best to get this information across to all readers.

Nice job overall!

Round 3 Reviews

Reviewer #1 (Remarks to the Author):

Thank you for reviewing the previous comments carefully, you have made nice improvements/corrections. Please see below for comments and suggestion on two of the remaining figures.

Figure 2: The off-center CCD images are beautiful and for the first time I can see the streaking that you are referring too. It would be my opinion that you use these images in the main text and leave the SAED images for the supplemental. Perhaps show the shortest exposure time SAED if you want to show the entire DP, but I think the CCD images show the streaking quite well. It might also be worth adding an additional arrow to help the reader locate it. As well, when thinking about publication, you might want to increase the contrast on the image and up the brightness so that the streaking is more apparent. You can also add indices so that the reader is aware of what reflection they are seeing in the off-axis image.

C1: At the reviewer's request, we have replaced Figure 2 with these CCD images, and added in crystallographic directions to aid the reader's understanding. Text has been updated accordingly as well. In addition, the films are shown in the SI with more thorough crystallographic notation. Due to how far off center the CCD images are, it was not trivial to determine the identity of the reflections themselves. Lastly, the images which were included in the previous reviewer's response already had the contrast adjusted to improve the visibility of the diffuse streaking, hence these same images are included in the new manuscript.

Figure 6: This is an improvement. It is not easy to glean information from the figure in the current format, however, it is sufficient, I would draw your attention to modern work using DPC imaging and Lorentz TEM/ STEM imaging. This same type of work is showing local changes in magnetic field, but it produces similar data to what you are trying to present in this figure. I highly recommend trying to color map your data to better get your point across. and provide a more elegant appearance. Please see the work presented for Fig 2,3,&4 in this paper by Shibata. DOI: 10.1038/ncomms15631, many other examples exist in various publications, please explore.

C2: We empathize with the reviewer (and the reader) that the figure is challenging to understand. However, we would contest that showing the vector displacement as a color map would further complicate this image, and would not be any clearer than the arrowheads, as only a 1-D vector is needed to be shown. In addition, by using the color map, one would not be able to show the overlaid model displacement structure. As the image stands, it demonstrates the experimental process, directly compares a model to the sample, and displays displacement vectors, all in a single image. We agree that there may be a better way to represent this data, but we are unaware of how, and would argue that a colormap would unnecessarily complicate the image. Lastly, we would reiterate that 6a is not the take away image, rather it is 6b, which shows the expected difference between the two row types.

For the record, I still have fundamental differences on the ability to use STEM imaging to make the statements you are claiming with regard to the small atomic position

displacements, but moving past that, your figures support the text. I would encourage you to think one last time about how best to get this information across to all readers.